# Phosphoproteomics identifies a bimodal EPHA2 receptor switch that promotes embryonic stem cell differentiation

Rosalia Fernandez-Alonso[1], Francisco Bustos [1], Manon Budzyk[1], Pankaj Kumar [2], Andreas O. Helbig[3], Jens Hukelmann[4], Angus I. Lamond[4], Fredrik Lanner [2], Houjiang Zhou[1], Evangelia Petsalaki [5] & Greg M. Findlay [1✉]

Embryonic Stem Cell (ESC) differentiation requires complex cell signalling network dynamics, although the key molecular events remain poorly understood. Here, we use phosphoproteomics to identify an FGF4-mediated phosphorylation switch centred upon the key Ephrin receptor EPHA2 in differentiating ESCs. We show that EPHA2 maintains pluripotency and restrains commitment by antagonising ERK1/2 signalling. Upon ESC differentiation, FGF4 utilises a bimodal strategy to disable EPHA2, which is accompanied by transcriptional induction of EFN ligands. Mechanistically, FGF4-ERK1/2-RSK signalling inhibits EPHA2 via Ser/Thr phosphorylation, whilst FGF4-ERK1/2 disrupts a core pluripotency transcriptional circuit required for *Epha2* gene expression. This system also operates in mouse and human embryos, where EPHA receptors are enriched in pluripotent cells whilst surrounding lineage-specified trophectoderm expresses EFNA ligands. Our data provide insight into function and regulation of EPH-EFN signalling in ESCs, and suggest that segregated EPH-EFN expression coordinates cell fate with compartmentalisation during early embryonic development.

[1] MRC Protein Phosphorylation and Ubiquitylation Unit, School of Life Sciences, University of Dundee, Dundee, UK. [2] Department of Clinical Science, Intervention and Technology, Ming Wai Lau Center for Reparative Medicine, Division of Obstetrics and Gynecology, Karolinska Institutet, 14186 Stockholm, Sweden. [3] Institute for Experimental Medicine, Christian Albrechts University, Kiel, Germany. [4] Centre for Gene Regulation and Expression, School of Life Sciences, University of Dundee, Dundee, UK. [5] European Molecular Biology Laboratory, European Bioinformatics Institute, Wellcome Genome Campus, Cambridge, UK. ✉email: g.m.findlay@dundee.ac.uk

Differentiation of pluripotent embryonic stem cells (ESCs) into specialised cell types requires remodelling of transcriptional and protein networks coupled to an ability to organise distinct cell populations[1]. This is driven by cellular responses to extracellular stimuli and activation of intracellular signalling networks. However, the key molecular changes that drive differentiation remain obscure. A systems-level view of developmental signalling is therefore required to comprehensively map fundamental mechanisms that drive pluripotent exit and acquisition of specialised cellular characteristics.

A critical signal driving differentiation of pluripotent cells is fibroblast growth factor 4 (FGF4)[2–6]. FGF4 acts via multiple signalling pathways, including ERK1/2, PI3K–AKT and PLCγ, and plays an overarching role in specification and organisation of early embryonic cell types[7,8]. Whilst FGF4 signalling profoundly modifies gene expression and cellular behaviour to promote differentiation[9,10], the molecular mechanisms by which the FGF4 network operates to drive differentiation remain poorly understood.

In this paper, we employ phosphoproteomic profiling to identify phosphorylation events by which FGF4 promotes exit from pluripotency towards differentiation. We identify the EPHA2 receptor tyrosine kinase as a critical target of the FGF4 signalling network in ESCs. EPH receptors engage transmembrane Ephrin ligands (EFNs)[11] to drive segregation of EPH- and EFN- expressing cell populations[12], and demarcation of distinct cellular compartments[13]. By exploring the function of EPH–EFN signalling in pluripotent cells, we show that EPHA2 is the primary mediator of EFN ligand responses in ESCs. Furthermore, activation of EPHA2 by EFNA1 supports pluripotency gene expression, and suppresses commitment by restraining ERK1/2 activation. During ESC differentiation, FGF4 signalling functionally disables EPHA2, which is accompanied by transcriptional induction of EFN ligands. Inhibition of EPHA2 occurs via a dual mechanism: FGF4–ERK1/2 signalling to ribosomal S6 kinase (RSK/RPS6K) drives inhibitory serine/threonine (S/T) phosphorylation of EPHA2, which blunts the response to EFN ligand. In parallel, FGF4–ERK1/2 extinguishes the pluripotency gene regulatory network to suppress *Epha2* expression and inactivate EPH–EFN signalling in differentiating ESCs. Our data thereby demonstrate a role for EPHA2 in pluripotency maintenance, and identify key transcriptional and post-translational mechanisms by which FGF4 signalling inhibits EPHA2 to reinforce the transition from pluripotency towards differentiation. Importantly, A-type EPH receptors are expressed in the pluripotent compartment of both human and mouse embryos, whilst the first specified lineage, trophectoderm, is enriched for *Efna1/EFNA1*, suggesting that segregated expression of A-type EPH receptors and EFN ligands plays a role in pluripotency maintenance during early embryonic development.

## Results

### EPHA2 is serine phosphorylated upon ESC differentiation. 
In order to comprehensively map potential mechanisms by which phosphorylation controls ESC differentiation, we conducted a phosphoproteomic survey of mouse ESC (mESC) signalling responses to the key differentiation factor FGF4 (Fig. 1a). To this end, we employ $Fgf4^{-/-}$ mESCs, which fail to activate FGF-dependent signalling pathways and transcription required for differentiation[4], and therefore remain pluripotent[14]. Importantly, providing $Fgf4^{-/-}$ mESCs with recombinant FGF4 restores signalling (Supplementary Fig. 1A) and differentiation responses[4,14].

We employed this inducible system to screen for FGF4-dependent phosphorylation sites following acute FGF4 stimulation using mass-spectrometry (MS)-based phosphoproteomic profiling. Although FGF4-dependent transcriptional responses require several hours, maximal activation of key downstream signalling pathways, such as ERK1/2 MAP kinase, occurs within minutes (Supplementary Fig. 1A)[14]. This suggests that proteins relevant for FGF4-dependent differentiation are likely to be phosphorylated within this time frame. Thus, control $Fgf4^{-/-}$ mESCs, or those stimulated with FGF4 for 5 or 20 min, were used to generate tryptic peptides. Phosphopeptides were enriched, labelled and fractionated using basic C18 reverse phase, and subjected to tandem MS (LC–MS/MS) (Fig. 1b). This analysis quantified 19,846 phosphopeptides, including 12,528 unique phosphosites on 3260 unique proteins. Of these, ~56% have not been previously identified (5404 were previously reported in PhosphositePlus[15]). The abundance of 2399 phosphopeptides is significantly altered at one or both time points of FGF4 stimulation (>2-fold change). This cohort includes known targets of the FGF4 pathway, including tyrosine phosphorylation of the scaffolding protein GAB1, and key components of the ERK1/2 MAP kinase pathway downstream of FGF4, such as SOS1, RAF1 or RPS6KA3 (Fig. 1c, d). Furthermore, this dataset greatly expands our understanding of the FGF signalling network in ESCs, when compared with previous large-scale phosphoproteomics datasets of FGF stimulation in ESCs[16,17] (Supplementary Fig. 1B).

As a means to pinpoint mechanisms by which FGF4 modulates intracellular signalling during differentiation, we focussed on the protein kinase superfamily. We find that FGF4 regulates phosphorylation of a significant panel of protein kinases, including pluripotency regulators RAF1, RPS6KA3[18] and poorly studied members of the protein kinase superfamily (Fig. 1e). Strikingly, EPHA2, a member of the Ephrin (EFN) receptor tyrosine kinase family, is phosphorylated at S898 in response to FGF4 stimulation (Fig. 1e). We confirm that this site is robustly phosphorylated over a time course of FGF4 stimulation using an EPHA2 S898 phosphospecific antibody (Fig. 1f), suggesting that EPHA2 is a bona fide target of FGF4 signalling during mESC differentiation.

### EPHA2 is the most abundant receptor kinase in mESCs. 
EPHA2 is a member of the large EPH receptor tyrosine kinase family[11], which prompted us to examine the expression profile of EPH receptor family members in mESCs. Quantitative total cell proteomics of around 10,000 proteins[19] reveals that EPHA2 is the major EPH receptor family member and the most abundant receptor kinase in mESCs (Fig. 2a). mESCs also express EPHA4, EPHB4, EPHB2 and EPHB3 at lower levels (Fig. 2a and Supplementary Fig. 2A), consistent with cell surface proteomics from mESCs and the early embryo[20]. Quantitative real-time polymerase chain reaction (qRT-PCR) analysis of members of the mammalian EPH receptor family (EPHA1–8, EPHB1–4 and EPHB6) confirms that *Epha2, Epha4, Ephb2, Ephb3* and *Ephb4* are expressed at the mRNA level in mESCs, along with *Epha1* and *Epha7* (Supplementary Fig. 2B). However, our data suggest that EPHA2 is the major EPH receptor expressed in mESCs, and may therefore play a key role in driving EPH–EFN signalling in these cells.

### EPHA2 is an essential mediator of EFN signalling in mESCs. 
In order to test the prediction that EPHA2 mediates EFN signalling responses in mESCs, we developed an unbiased functional assay to identify EPH receptor family members that engage EFN ligand in cell extracts. This approach employs EFNA1 and B1 as representatives of the EFNA and EFNB classes, which possess the capacity to promiscuously engage EPHA and EPHB family members, respectively[21]. EPH family members captured using recombinant EFNA1 and B1 from mESC lysates were detected and quantified using affinity purification MS (Fig. 2b). Proof of

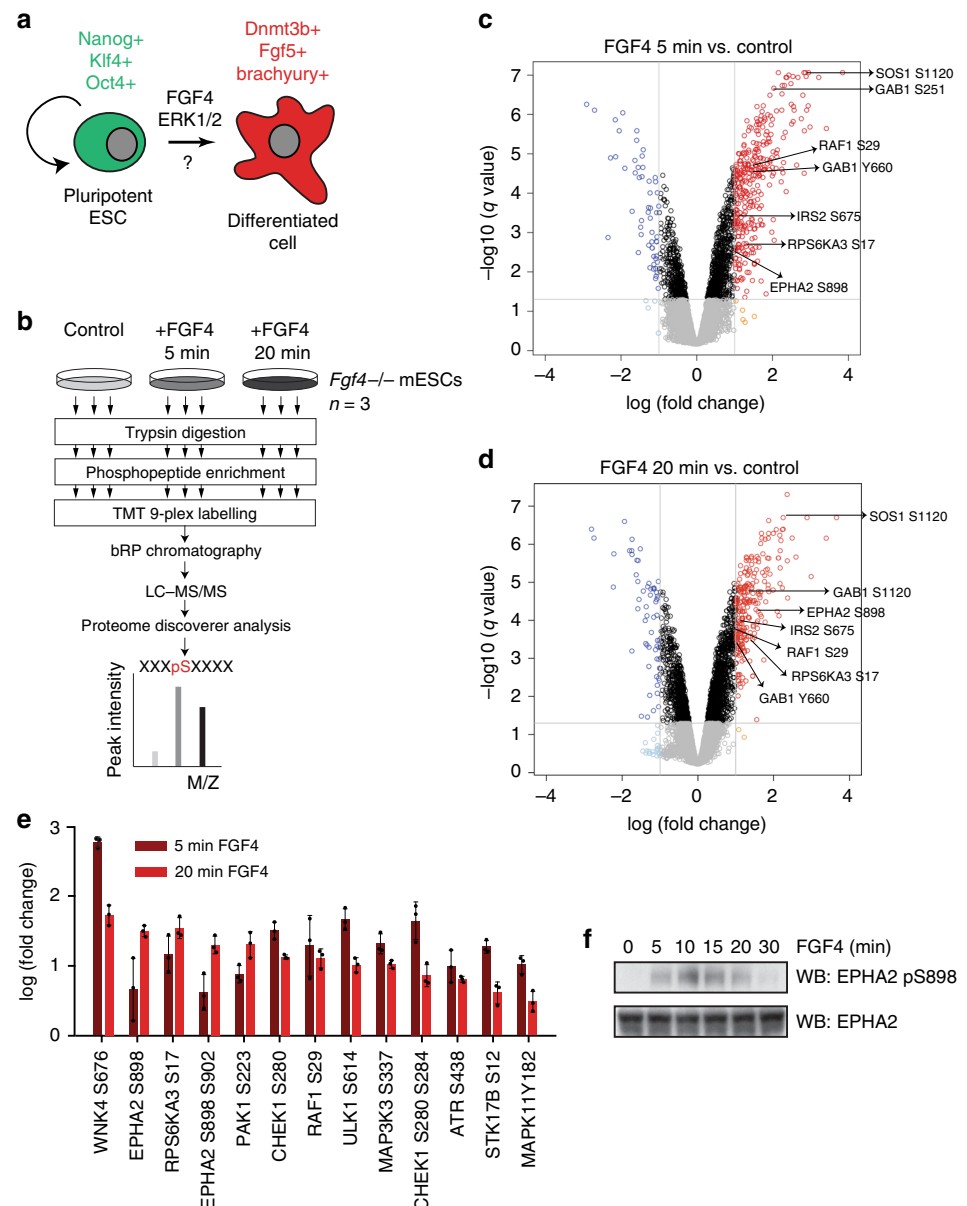

**Fig. 1 FGF4 signalling in mESCs promotes EPHA2 Ser phosphorylation. a** FGF4 is a key signal that promotes differentiation of pluripotent mESCs. **b** Workflow for phosphoproteomic analysis of FGF4 signalling in $Fgf4^{-/-}$ mESCs. Volcano plot showing significantly modified phosphosites after stimulation of $Fgf4^{-/-}$ mESCs with FGF4 for 5 min (**c**) and 20 min (**d**). Phosphosites on known FGF4 pathway components are highlighted. **e** Protein kinase phosphopeptides that are significantly upregulated (>2-fold) on at least one time point (5 or 20 min) compared with control. Data are presented as mean ± SD ($n = 3$). **f** $Fgf4^{-/-}$ mESCs were stimulated with FGF4 for the indicated time, and EPHA2 pS898 and EPHA2 levels determined by immunoblotting. Source data are provided as a Source Data file.

concept for this method was established by immunoblotting for EPHA2, which is effectively captured by EFN pulldown (Fig. 2c). Coomassie staining of EFNA1–B1 pulldowns confirms the capture of EFNA1 and B1, and reveals further specific bands in the 75–130-kDa molecular weight range (Fig. 2d). MS analysis of these samples primarily identified EPHA2 associated specifically with EFNA1, with few peptides identified from other EPH family members (Fig. 2e). Therefore, although mESCs express several EPH receptor family members, EPHA2 is the major family member that engages EFN ligands.

To test the hypothesis that EPHA2 is a key regulator of EFN signalling in mESCs, we generated $Epha2^{-/-}$ mESCs using CRISPR/Cas9, and examined tyrosine-phosphorylated proteins that interact with EFNA1–B1 following stimulation of mESCs. EFNA1–B1 captures a single tyrosine-phosphorylated protein of

~100 kDa in $Epha2^{+/+}$ mESCs (Fig. 2f). However, this is lost in $Epha2^{-/-}$ mESCs (Fig. 2f), indicating that specific disruption of EPHA2 expression abolishes the phosphotyrosine response to EFNA1–B1 stimulation. These data suggest that EPHA2 is the major receptor that mediates EFN signalling responses in mESCs. However, we observe a moderate increase of $Epha1$ expression in three $Epha2^{-/-}$ mESCs clonal cell lines (Supplementary Fig. 2C), suggesting that compensatory upregulation[22,23] in early embryonic cells may mask embryonic $Epha2^{-/-}$ phenotypes.

**EPHA2 activation by EFNA1 occurs in *trans* in mESCs.** Although EPHA2 can be activated by EFNA1-recombinant ligand in mESCs (Fig. 2f), physiological EPH receptor activation occurs via cell surface-expressed EFNs. Therefore, we set out to determine

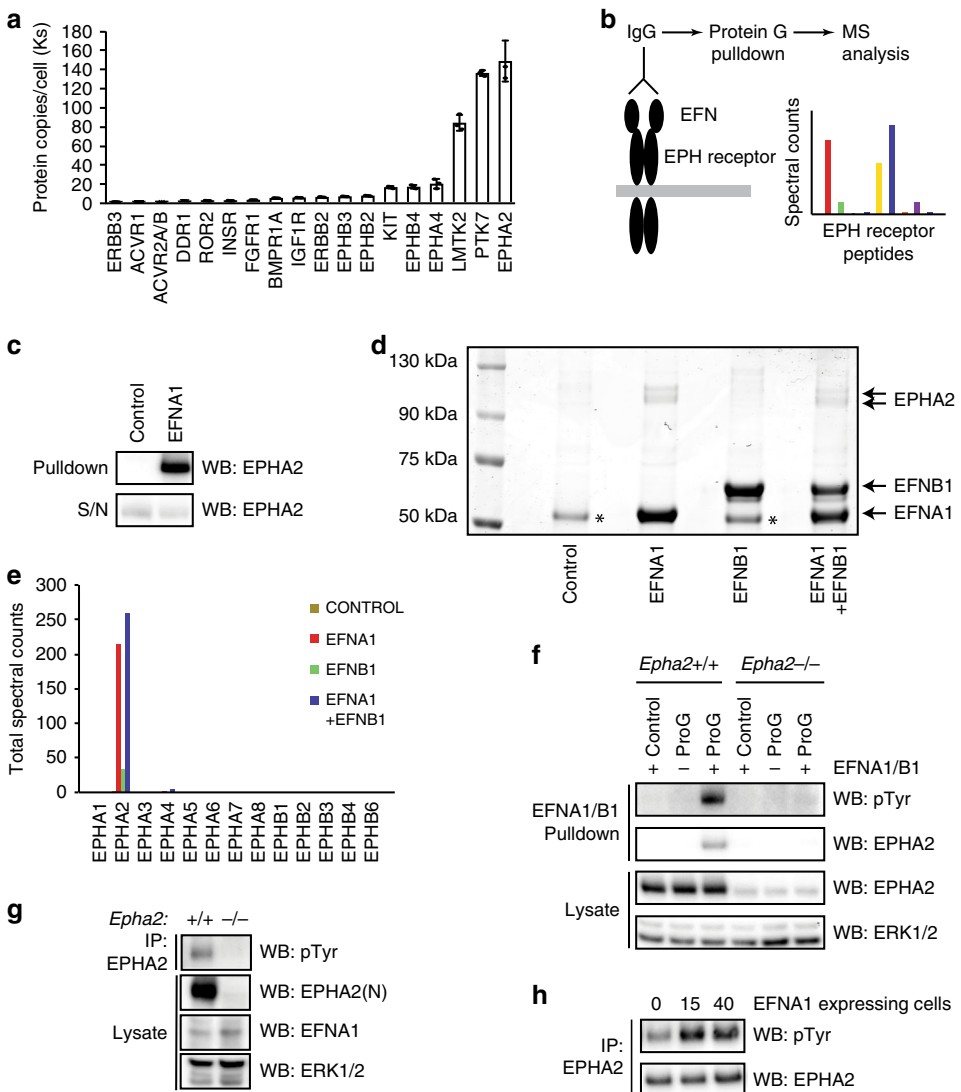

**Fig. 2 EPHA2 is critical for EFN ligand responses in mESCs. a** Average protein copy number per cell determined for receptor kinases in mESCs, using quantitative whole-cell proteomics. Data are presented as mean ± SD (*n* = 3). **b** Workflow for quantification of EPH–EFN interactions in mESCs by EFN ligand affinity purification mass spectrometry. **c** Proof-of-principle identification of EPH–EFN interactions by EFN ligand affinity purification. EPHA2 levels were determined by immunoblotting. **d** Coomassie staining of EFNA1/B1 affinity purification from mESCs. EFNA1, EFNB1 and EPHA2 proteins are indicated. (*) = non-specific band. **e** Mass-spectrometry analysis of 75–130-kDa region of the Coomassie stained EFNA1/EFNB1 affinity purification shown in (**d**). Total spectral counts recovered for each EPH receptor family member are indicated. **f** EFNA1/B1 affinity purification from intact *Epha2+/+* and pooled *Epha2−/−* mESCs. Phosphotyrosine (pTyr), EPHA2 and ERK1/2 levels were determined by immunoblotting. Note that the pTyr signal is specific for EPHA2, and is not detected in the absence of EFN ligand. **g** EPHA2 was immunoprecipitated from *Epha2+/+* and *Epha2−/−* mESCs, and pTyr, EPHA2, EFNA1 and ERK1/2 levels determined by immunoblotting. **h** *Epha2+/+* mESCs were stimulated with EFNA1-expressing *Epha2−/−* mESCs for 15 or 40 min. EPHA2 was immunoprecipitated, and pTyr and EPHA2 levels determined by immunoblotting. Source data are provided as a Source Data file.

how EPHA2 is activated in mESCs. Interestingly, EPHA2 is tyrosine phosphorylated in mESCs under standard leukaemia-inhibitory factor (LIF) and foetal bovine serum (FBS) culture conditions (Fig. 2g), which may be attributable to expression of A-type EFN ligands such as EFNA1 in mESCs. Indeed, EFNA1-expressing *Epha2−/−* mESCs robustly stimulate EPHA2 activation in *Epha2+/+* mESCs following short-term co-culture (Fig. 2h), indicating that EFNA1 drives EPHA2 activation in *trans*. This is consistent with previous data reporting that *trans* interactions between EPH receptors and EFN ligands activate signalling, whilst *cis* interactions are inhibitory[24–26].

**EPH–EFN signalling supports pluripotency factor expression.** As EPH–EFN signalling is active in cultured mESCs, we next sought to determine the function of this pathway in these cells.

To this end, we tested whether EPH–EFN signalling plays a role in regulation of mESC pluripotency and/or differentiation. We generated multiple *Epha2−/−* mESC clones and an isogenic *Epha2−/−* mESC clone in which EPHA2 expression is stably reintroduced at endogenous levels. Firstly, we used these mESC lines to investigate the role of EPHA2 in controlling expression of pluripotency gene regulatory network components. When cultured in LIF/FBS conditions, *Epha2−/−* mESCs express similar levels of pluripotency factors such as NANOG, KLF4 and OCT4, and the differentiation marker DNMT3B (Supplementary Fig. 3A). However, following mESC differentiation initiated by 48 h of LIF withdrawal, *Epha2−/−* mESCs display reduced KLF4 and increased DNMT3B expression, when compared with control *Epha2+/+* mESCs (Fig. 3a). This effect is rescued by EPHA2 expression in *Epha2−/−* mESCs (Fig. 3a).

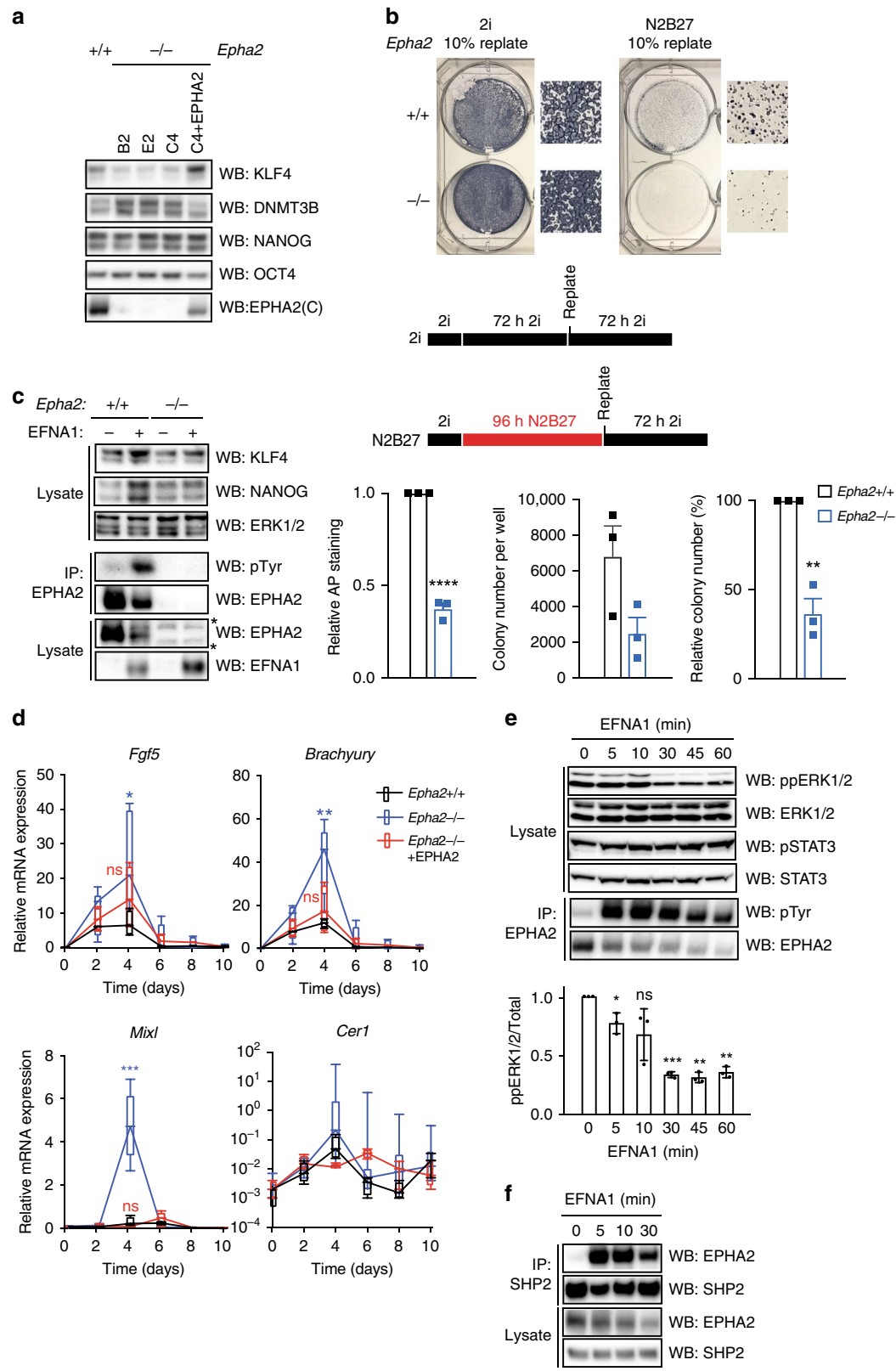

Thus, whilst EPHA2 is not required for pluripotency under standard mESC culture conditions, EPHA2 supports expression of a subset of pluripotency genes upon LIF withdrawal. Importantly, EPHA2 signalling has no effect on mESC proliferation (Supplementary Fig. 3B, C).

**EPH–EFN signalling restricts commitment to differentiation.** In order to directly determine whether EPHA2 restrains exit from pluripotency, we performed commitment assays on mESCs cultured in 2i media[27]. Epha2[+/+] or Epha2[−/−] mESCs were differentiated in N2B27 medium for 4 days, then replated in 2i medium

**Fig. 3 EFNA1-activated EPHA2 supports mESC pluripotency and restricts commitment to differentiation by suppressing ERK1/2 signalling. a** $Epha2^{+/+}$, $Epha2^{-/-}$ or $Epha2^{-/-}$ mESCs (clone C4) stably expressing EPHA2 were cultured in the absence of LIF for 48 h. EPHA2, KLF4, DNMT3B, NANOG and OCT4 levels were determined by immunoblotting. **b** $Epha2^{+/+}$ or $Epha2^{-/-}$ (clone C4) mESCs were maintained in 2i or differentiated in N2B27 media for 72 or 96 h, respectively, whereupon 10% of cells were replated in 2i. Total alkaline phosphatase staining is represented relative to $Epha2^{+/+}$ mESCs. The total number of alkaline phosphatase-positive colonies for $Epha2^{+/+}$ and $Epha2^{-/-}$ mESCs is shown, and also represented relative to $Epha2^{+/+}$ mESCs. Data show mean ± SEM ($n = 3$); statistical significance was determined using unpaired two-sided Student's $t$ test comparing $Epha2^{-/-}$ with the $Epha2^{+/+}$ control (****$P < 0.0001$, **$P = 0.0016$). **c** $Epha2^{+/+}$ or $Epha2^{-/-}$ (clone C4) mESCs stably expressing EFNA1, along with the respective parental controls, were grown in LIF/FBS, and KLF4, NANOG, EPHA2, EFNA1 and ERK1/2 levels determined by immunoblotting, or EPHA2 immunoprecipitated and pTyr and EPHA2 levels determined by immunoblotting. **d** $Epha2^{+/+}$, $Epha2^{-/-}$ or $Epha2^{-/-}$ mESCs stably expressing EPHA2 were differentiated as embryoid bodies for 10 days, and the levels of $Fgf5$, $Brachyury$, $Mixl$ and $Cer1$ mRNA determined by qRT-PCR. Box-and-whisker plots show median, first and third quartiles, and maximum and minimum values. The results shown are for technical replicates from two independent experiments, including three $Epha2^{-/-}$ clones ($n = 3$); statistical significance at day 4 was determined using unpaired two-sided Student's $t$ test comparing each group with the $Epha2^{+/+}$ control (ns = not significant, *$P = 0.0252$, **$P = 0.0045$, ***$P < 0.0001$). **e** $Epha2^{+/+}$ mESCs cultured in LIF/FBS were stimulated with 1 μg/ml clustered EFNA1 for the indicated times. ppERK1/2, total ERK1/2, STAT3 pY705 and total STAT3 levels were determined by immunoblotting. EPHA2 was immunoprecipitated, and pTyr and EPHA2 levels determined by immunoblotting. ppERK1/2 signal was quantified; data show mean ± SD ($n = 3$); statistical significance was determined using one-sample two-sided $t$ test comparing each group with control, theoretical mean = 1 (ns = not significant, 5 min; *$P = 0.0467$, 30 min; ***$P = 0.0005$, 45 min; **$P = 0.0014$, 60 min; **$P = 0.0018$). **f** $Epha2^{+/+}$ mESCs cultured in LIF/FBS were stimulated with 1 μg/ml clustered EFNA1 for the indicated times. SHP2 was immunoprecipitated, and SHP2 and EPHA2 levels detected by immunoblotting. Source data are provided as a Source Data file.

and colonies assessed for alkaline phosphatase activity. As expected, most cells commit to differentiation during N2B27 culture (Supplementary Fig. 3D). However, the number of alkaline phosphatase-positive colonies recovered from $Epha2^{+/+}$ was significantly higher than that from $Epha2^{-/-}$ mESCs (Fig. 3b), indicating that EPHA2 suppresses mESC commitment to differentiation.

We also investigated the impact of activating EPH–EFN signalling on mESC pluripotency maintenance. To this end, we generated mESCs stably expressing the EFN ligand EFNA1, in either an $Epha2^{+/+}$ or $Epha2^{-/-}$ background. EFNA1-expressing $Epha2^{+/+}$ mESCs display an increase in EPHA2 tyrosine phosphorylation, which correlates with elevated expression of pluripotency markers NANOG and KLF4 (Fig. 3c). However, expression of EFNA1 in $Epha2^{-/-}$ cells does not alter NANOG and KLF4 expression (Fig. 3c), indicating that this effect occurs via EPHA2 activation. In a similar vein, EFNA1-expressing $Epha2^{+/+}$ mESCs show strong alkaline phosphatase staining and rounded colony morphology following mESC commitment assay, in contrast to EFNA1-expressing $Epha2^{-/-}$ mESCs (Supplementary Fig. 3E). Taken together, our data indicate that EPHA2 activation by EFNA1 supports key mESC morphological and transcriptional characteristics, thereby uncovering a function for EPH–EFN signalling in mESC pluripotency.

**EPHA2 suppresses expression of differentiation markers**. As our data suggest that EPHA2 may function to restrict mESC differentiation, we investigated the effects of EPHA2 in an embryoid body (EB) differentiation model, which mimics elements of early embryonic development. EB aggregation effectively promotes multi-lineage differentiation, including $Fgf5^+$ epiblast, and $Brachyury^+$, $Mixl^+$ mesendoderm and $Cer1^+$ endoderm (Fig. 3d). This is accompanied by a concomitant decrease in expression of pluripotency markers $Nanog$, $Klf4$ and $Oct4$, and induction of the differentiation marker $Dnmt3b$ (Supplementary Fig. 3F). Strikingly, $Epha2^{-/-}$ mESCs display augmented expression of $Fgf5$, $Brachyury$, $Mixl$ and to a lesser extent $Cer1$ following EB differentiation, which is restored by EPHA2 re-expression in $Epha2^{-/-}$ EBs (Fig. 3d), consistent with a role for EPHA2 in suppressing mESC differentiation.

We also explored the function of EPHA2 in controlling pluripotency and differentiation when mESCs are maintained in basal media containing the MEK1/2 inhibitor PD0325901 and GSK3 inhibitor CHIR99021 (2i) to promote 'ground state' naive pluripotency prior to differentiation in N2B27 media. Again,

decay of pluripotency factors $Nanog$, $Klf4$ and $Oct4$ is not significantly altered in $Epha2^{-/-}$ mESCs, although expression of the early differentiation factor $Dnmt3b$ is elevated in these cells (Supplementary Fig. 3G). $Epha2^{-/-}$ mESCs also display significantly elevated expression of the neural-specific markers $Sox1$ and $Nestin$ following N2B27 differentiation (Supplementary Fig. 3G). Expression of the axonal transport factor $Kif1a$ shows an upward trend, although this is not statistically significant (Supplementary Fig. 3G). We also assessed expression of the mesendoderm marker $Brachyury$ in the 2i/N2B27 differentiation system. However, $Brachyury$ levels are high in 2i media due to transcriptional activation observed upon Wnt pathway activation/GSK3 inhibition (Supplementary Fig. 3G). Nevertheless, data from the mESC commitment assay (Fig. 3b) and EB differentiation system (Fig. 3d) indicate that EPHA2 restricts mESC differentiation in distinct models, consistent with a general role of EPHA2 in regulating commitment to differentiation. However, further experiments are required to definitively elucidate the function of EPHA2 in lineage specification.

**EPHA2 activation antagonises ERK1/2 signalling in mESCs**. Our findings prompted us to determine the mechanism by which EPHA2 supports mESC pluripotency and restricts commitment. As the LIF–JAK–STAT3 and FGF4–ERK1/2 signalling pathways play key roles in mESC pluripotency and differentiation, respectively, we hypothesised that EPHA2 activation modulates signalling via one or both of these pathways. Stimulation of mESCs with clustered recombinant EFNA1 activates EPHA2, as measured by EPHA2 tyrosine phosphorylation (Fig. 3e). Intriguingly, EPHA2 activation specifically suppresses ERK1/2 activation without affecting signalling via the LIF–JAK–STAT3 pathway (Fig. 3e). This is consistent with a previously described role for EPHA2 in inhibiting ERK1/2 activation in other cellular systems[28,29], and suggests that EPHA2 restrains differentiation via specific inhibition of ERK1/2.

Next, we investigated the proposed interaction between EFN–EPH and ERK1/2 signalling pathways. FGF4–ERK1/2 inhibition promotes pluripotency gene expression even in $Epha2^{-/-}$ mESCs, reversing the effects of $Epha2$ gene knockout on DNMT3B and KLF4 expression (Supplementary Fig. 3H). We have shown previously that addition of exogenous FGF4 to $Fgf4^{-/-}$ mESCs elevates DNMT3B and suppresses KLF4 expression[14], thereby phenocopying $Epha2$ gene knockout. Furthermore, similar to FGF4–ERK1/2 inhibition, EPHA2 activation drives expression of NANOG (Fig. 3c), reinforcing the functional interaction between

EFN–EPH and ERK1/2 signalling in regulating pluripotency gene expression. In summary, our data indicate that EFN–EPH and FGF4–ERK1/2 have broadly opposing functions in regulating expression of pluripotency gene network components. However, it should be noted that NANOG expression is not significantly altered in $Epha2^{-/-}$ mESCs (Fig. 3a).

To provide further mechanistic insight, we sought to determine how EPHA2 intersects with the ERK1/2 pathway. SHP2/PTPN11 is a protein tyrosine phosphatase that provides a key scaffold[14] to promote ERK1/2 activation and ESC differentiation[30–33]. As EPHA2 has been shown to interact with SHP2[34], we tested whether EPHA2 engages SHP2 in mESCs. Indeed, EFNA1 stimulation drives recruitment of SHP2 to EPHA2 (Fig. 3f), suggesting that EPHA2 activation inhibits ERK1/2, either by sequestering the key scaffold SHP2, or by recruiting SHP2 to dephosphorylate key phosphotyrosine sites that are required for ERK1/2 activation.

**FGF4–RSK signalling promotes EPHA2 S/T phosphorylation.** As demonstrated, FGF4-dependent differentiation signalling promotes EPHA2 S898 phosphorylation (Figs. 1f, 4a). This prompted us to investigate the regulation and function of EPHA2 S898 phosphorylation in mESCs. Previous studies have implicated several kinases in phosphorylation of the EPHA2 S898 motif, including the AGC family kinase AKT, p90 ribosomal S6 kinase (RSK/RPS6K) and PKA[35–37]. FGF4-dependent EPHA2 S898 phosphorylation in mESCs is blocked by the FGFR inhibitor AZD4547 and MEK1/2 inhibitor PD0325901 (Fig. 4b), confirming a key role for the FGF4–ERK1/2 MAP kinase pathway. As RSK is phosphorylated and activated by ERK1/2[38], we hypothesised that RSK is the EPHA2 S898 kinase in mESCs. Indeed, structurally distinct RSK inhibitors BI-D1870 and SB747651A block FGF4-dependent EPHA2 S898 phosphorylation (Fig. 4b). Importantly, these inhibitors do not consistently inhibit AKT activation (Fig. 4b), whilst EPHA2 S898 phosphorylation is insensitive to PI3K inhibitors Wortmannin and LY294002 (Fig. 4b). Therefore, our data define the FGF4–ERK1/2–RSK pathway, rather than PI3K–AKT, as the major driver of EPHA2 S898 phosphorylation in response to FGF4 in mESCs.

S898 lies in the intracellular region of EPHA2 proximal to the kinase domain. Within this motif lie four further S/T residues previously identified by MS analysis (Fig. 4a, www.phosphositeplus.org)[37]. We therefore examined phosphorylation of this motif in FGF4-stimulated mESCs by MS. Analysis of endogenous EPHA2 immunoprecipitated from $Fgf4^{-/-}$ mESCs stimulated with FGF4 and EFNA1 reveals a tryptic peptide containing the S898 motif. This motif is phosphorylated on at least three S/T residues (Supplementary Table 1), although it was not possible to discern the exact positions by this method. Nevertheless, these data suggest extensive phosphorylation of the EPHA2 S898 motif in mESCs.

**EPHA2 S/T phosphorylation inhibits activation.** We then explored the role of S898 motif phosphorylation in EPHA2 regulation. To this end, we exploited $Epha2^{-/-}$ mESCs reconstituted with approximately endogenous EPHA2 expression levels to examine EPHA2 activation under conditions where signalling is acutely responsive to EFNA1 ligand stimulation (Supplementary Fig. 4A). Wild-type EPHA2 is efficiently activated by EFNA1 ligand, as measured by tyrosine phosphorylation (Fig. 4c and Supplementary Fig. 4B). However, phosphomimetic mutation of the five S/T sites within the S898 motif (5E) significantly inhibits EPHA2 activation by clustered EFNA1 (Fig. 4c and Supplementary Fig. 4B). These data suggest that multisite S/T phosphorylation at

the S898 motif disrupts ligand-induced EPHA2 tyrosine kinase activation.

In order to test this directly, we generated knock-in (KI) mESC lines expressing wild-type EPHA2 (WT KI) or a non-phosphorylatable mutant at the five S/T sites within the S898 motif (5A KI). Consistent with the inhibitory role of S898 motif phosphorylation on EPHA2 activation, EPHA2 5A is more active than wild-type EPHA2 (Fig. 4d). This result prompted us to examine the phenotype of EPHA2 5A KI mESCs. Strikingly, EPHA2 5A KI mESC colonies display a rounded morphology (Fig 4e), characteristic of ground-state pluripotency[39], which also resembles the morphology of mESCs in which EPHA2 is activated by EFNA1 (Supplementary Fig. 3E). Furthermore, EPHA2 5A KI mESCs show elevated expression of pluripotency markers NANOG and OCT4, which is accompanied by reduced expression of the differentiation marker DNMT3B (Fig. 4f). These data support the notion that S898 motif phosphorylation inhibits EPHA2 activation, and suppresses key morphological and transcriptional characteristics associated with pluripotent mESCs.

**FGF4 signalling suppresses EPHA2 during differentiation.** Given that FGF4 signalling dynamically remodels gene expression during mESC differentiation[9], we next investigated whether FGF4 might also control expression of EPH receptor genes. We employed EB aggregation as an FGF4–ERK1/2-dependent differentiation model (Supplementary Fig. 4C) to explore EPH–EFN gene expression dynamics. $Epha2$ and $Epha4$ mRNAs are substantially downregulated by day 6 (Fig. 4g; Supplementary Fig. 4D), which closely mirrors pluripotency factor suppression (Supplementary Fig. 3F). In contrast, EFN expression increases upon mESC differentiation (Fig. 4h), suggesting that FGF4 suppresses EPH receptor gene expression whilst driving increased expression of EFN ligands. Indeed, differentiation of $Fgf4^{-/-}$ mESCs in response to recombinant FGF4 similarly suppresses EPHA2 expression (Supplementary Fig. 4E).

We also investigated whether EPH and EFN gene expression is responsive to FGF4 signalling during differentiation in the 2i mESC model[39]. Indeed, removal of 2i activates FGF4–ERK1/2 signalling, leading to similar suppression of $Epha2$ mRNA (Fig. 4i) and EPHA2 protein (Supplementary Fig. 4F), with a corresponding increase in $Efna1$ mRNA (Fig. 4j). EPHA2 suppression is reversed by treatment of differentiating 2i mESCs with FGFR inhibitor AZD4547 or MEK1/2 inhibitor PD0325901, but not with the RSK inhibitor BI-D1870 or the PI3K inhibitor Wortmannin (Supplementary Fig. 4G). These results provide multiple lines of evidence that FGF4–ERK1/2 activity inhibits $Epha2$ gene transcription. However, unlike inhibitory EPHA2 S898 phosphorylation, this appears to be regulated independent of RSK kinase activity.

**The pluripotency gene network controls $Epha2$ expression.** Our results thus far suggest a close correlation between expression of EPHA2 and pluripotency factors (Fig. 4g, i, Supplementary Fig. 4E–G). We therefore tested whether the master pluripotency transcription factor OCT4[40] governs EPHA2 expression. OCT4 siRNA knockdown specifically suppresses EPHA2 expression (Fig. 5a), in contrast to knockdown of the SOX2 pluripotency transcription factor (Fig. 5a), which shares many targets with OCT4[41], and has a similar impact on NANOG/DNMT3B pluripotency signature (Supplementary Fig. 5A). Similarly, NANOG knockdown does not impact on EPHA2 expression (Supplementary Fig. 5B), and suppression of EPHA2 expression by the BRD4 inhibitor JQ1 correlates with loss of OCT4, but not NANOG

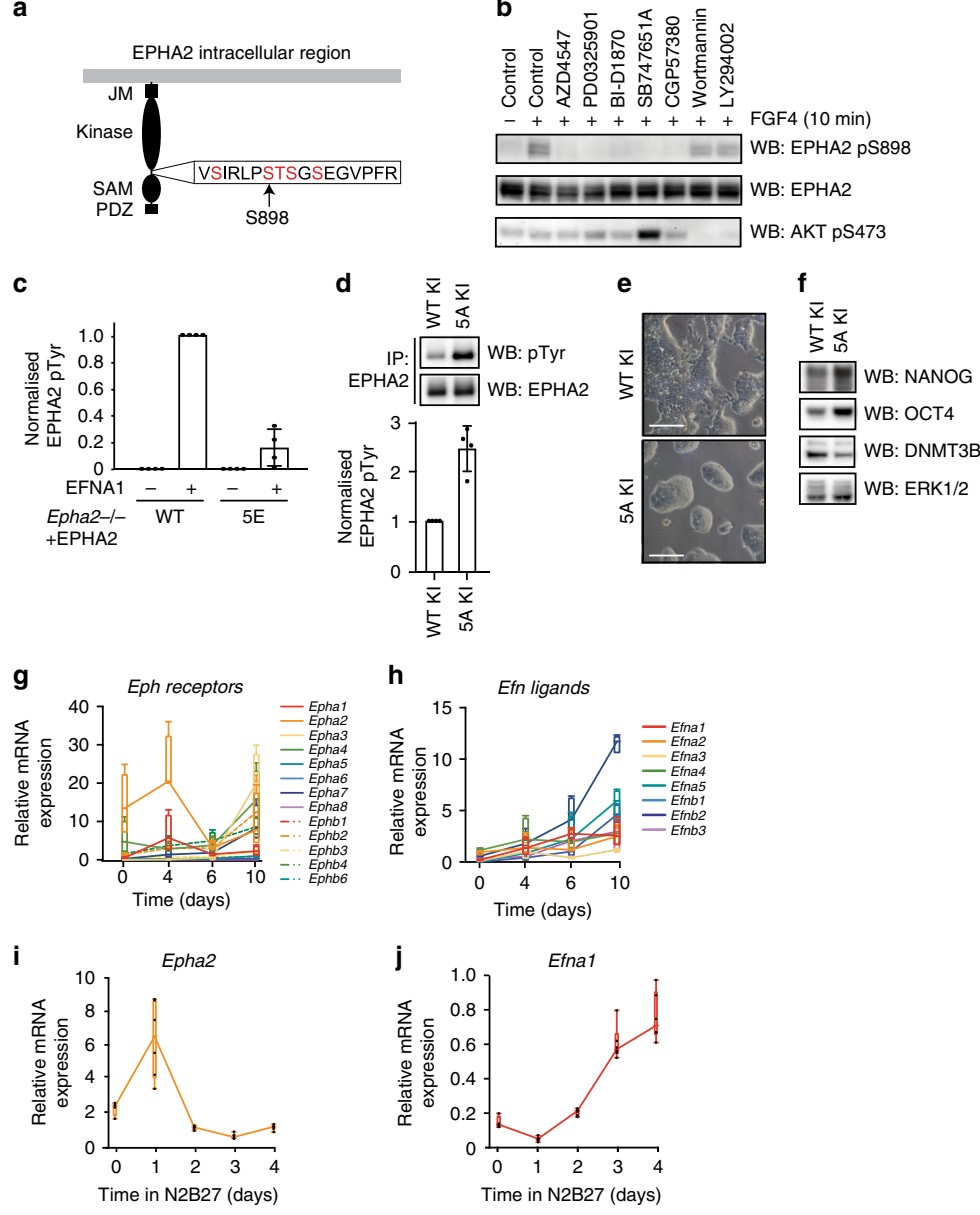

**Fig. 4 FGF4–ERK1/2 signalling inhibits EPHA2 activation and rewires EPH–EFN expression. a** Diagram of potential phosphorylation sites within the EPHA2 S898 motif. Mass spectrometry analysis detects phosphorylation of at least three sites in EPHA2 immunoprecipitated from FGF4-stimulated *Fgf4*−/− mESCs (see Supplementary Table 1). **b** *Fgf4*−/− mESCs were treated with 10 µM of the indicated inhibitors for 1 h, and stimulated with FGF4 for 10 min. EPHA2 pS898, EPHA2 and AKT pS473 levels were determined by immunoblotting. **c** *Epha2*−/− mESCs were transfected with either wild type or 5E EPHA2 constructs, and stimulated with 1 µg/ml clustered EFNA1 for 15 min. EPHA2 was immunoprecipitated, and pTyr and EPHA2 levels determined by immunoblotting and quantified. Data show mean ± SD (*n* = 4). **d** EPHA2 was immunoprecipitated from EPHA2 WT knock-in (KI) or 5A KI cell lines and pTyr and EPHA2 levels determined by immunoblotting (upper panel). Relative pTyr/EPHA2 signal was quantified (lower panel). Data show mean ± SD (*n* = 4). **e** Phase-contrast images of EPHA2 WT KI or 5A KI mESC lines; scale bar = 100 µM. **f** EPHA2 WT KI or 5A KI cell lines were cultured in LIF/FBS medium for 48 h, and KLF4, NANOG, DNMT3B, OCT4 ppERK1/2 and ERK1/2 levels determined by immunoblotting. **g** *Epha2*+/+ mESCs were differentiated as embryoid bodies for 10 days, and EPH receptor expression determined by qRT-PCR at the indicated time points. Box-and-whisker plots show median, first and third quartiles, and maximum and minimum values of four technical replicates (*n* = 4). **h** *Epha2*+/+ mESCs were differentiated as embryoid bodies for 10 days, and EFN ligand expression determined by qRT-PCR at the indicated time points. Box-and-whisker plots show median, first and third quartiles, and maximum and minimum values of four technical replicates (*n* = 4). *Epha2* (**i**) and *Efna1* (**j**) mRNA expression in 2i mESCs undergoing differentiation in N2B27 was determined by qRT-PCR analysis at the indicated time points. Box-and-whisker plots show median, first and third quartiles, and maximum and minimum values of two technical and three biological replicates (*n* = 3). Source data are provided as a Source Data file.

(Supplementary Fig. 5C). Thus, our results support a major role for OCT4 in regulating EPHA2 expression. However, it has been previously reported that loss of SOX2 leads to reduced *Oct4* expression[42], and we cannot rule out the possibility that residual

SOX2 and/or NANOG expression is sufficient to sustain expression of EPHA2.

In principle, EPHA2 suppression upon OCT4 depletion could occur as an indirect consequence of mESC differentiation. To test

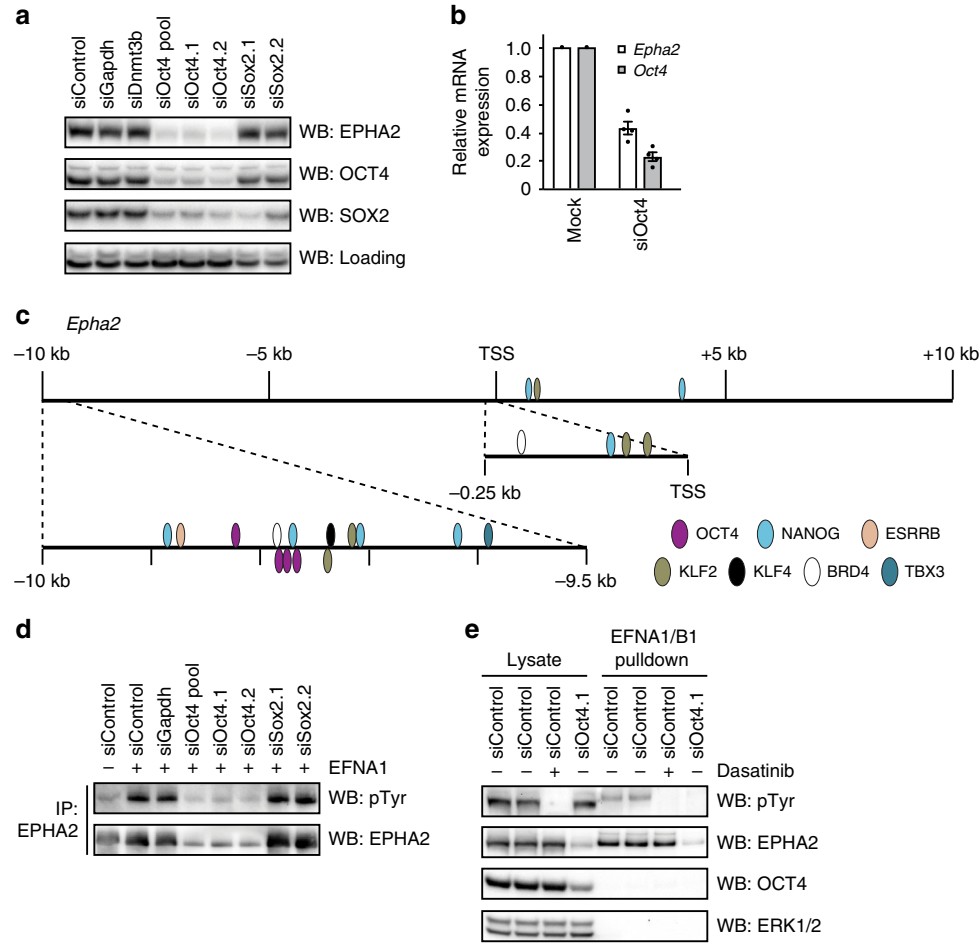

**Fig. 5 OCT4 is required for EPHA2 expression and EFN ligand responses in mESCs. a** $Epha2^{+/+}$ mESCs were transfected with the indicated siRNAs, and EPHA2, OCT4 and SOX2 levels determined by immunoblotting. A non-specific band was used as a loading control. **b** $Epha2$ and $Oct4$ mRNA expression was determined by qRT-PCR following transfection of $Epha2^{+/+}$ mESCs with control or OCT4 siRNA. Data show mean ± SEM of four technical replicates (n = 4). **c** Pluripotency transcription factor-binding sites in $Epha2$ gene regulatory regions were extracted from CODEX mESC ChIP-SEQ data (http://codex.stemcells.cam.ac.uk). **d** $Epha2^{+/+}$ mESCs were transfected with the indicated siRNAs, and stimulated with 1 μg/ml clustered EFNA1 for 15 min. EPHA2 was immunoprecipitated using EPHA2 antibody, and pTyr and EPHA2 levels determined by immunoblotting. **e** EFNA1/B1 affinity purification from control, dasatinib-treated or siOct4-transfected mESCs. pTyr, EPHA2, OCT4 and ERK1/2 levels were determined by immunoblotting. Source data are provided as a Source Data file.

this possibility, we performed acute (24 h) depletion of OCT4 by siRNA, which suppresses EPHA2 expression without affecting expression of pluripotency factors NANOG and SOX2 (Supplementary Fig. 5D). These data confirm that EPHA2 loss following OCT4 knockdown is not explained by altered mESC identity. siRNA knockdown of OCT4 also suppresses $Epha2$ mRNA levels (Fig. 5b), consistent with a role for OCT4 in directly regulating $Epha2$ gene transcription. Consistent with this, $Epha2$ gene regulatory elements form a hub for recruitment for OCT4 and other pluripotency transcription factors, suggesting that the pluripotency gene regulatory network directly controls $Epha2$ gene transcription in mESCs (Fig. 5c, http://codex.stemcells.cam.ac.uk). Interestingly, EPHA2 is expressed in a homogeneous manner characteristic of OCT4 (Supplementary Fig. 5E), rather than the heterogeneous expression pattern characteristic of other pluripotency factors such as NANOG and KLF4. Taken together, our data are consistent with a critical and specific function for OCT4 in regulating EPHA2 expression, although other pluripotency factors may also play a role.

**An OCT4-$Epha2$ transcriptional module controls EPH signalling.** Our results suggest that OCT4 may be an essential factor

for EPH–EFN signalling in pluripotent mESCs. We investigated this possibility by testing EFN signalling responses upon OCT4 siRNA knockdown. Following EFNA1 stimulation, EPHA2 immunoprecipitated from mESCs is robustly activated, as measured by tyrosine phosphorylation (Fig. 5d). However, mESCs transfected with OCT4 siRNA display loss of EPHA2 tyrosine phosphorylation in response to EFNA1 stimulation, consistent with an overall reduction in EPHA2 expression (Fig. 5d). More importantly, OCT4 suppression abolishes all EFNA1–B1-induced EPH receptor tyrosine phosphorylation (Fig. 5e). This signal is specific to tyrosine-phosphorylated EPHA2, as it is abolished by EPHA2 gene disruption (Fig. 2f) or treatment with the broad-spectrum tyrosine kinase inhibitor dasatinib (Fig. 5e). These data establish a critical function for OCT4 in enabling EPH receptor activation in pluripotent cells.

**EPH–EFN regulation in early embryonic cells.** Finally, we investigated whether the EPH–EFN signalling system is regulated in a similar manner in the early embryo. Single-cell RNA-sequencing data from distinct stages of early embryonic development[43] suggest that EPH receptors are not expressed in the pluripotent cells of the morula (embryonic day 2.5) (Supplementary Fig. 6), and are

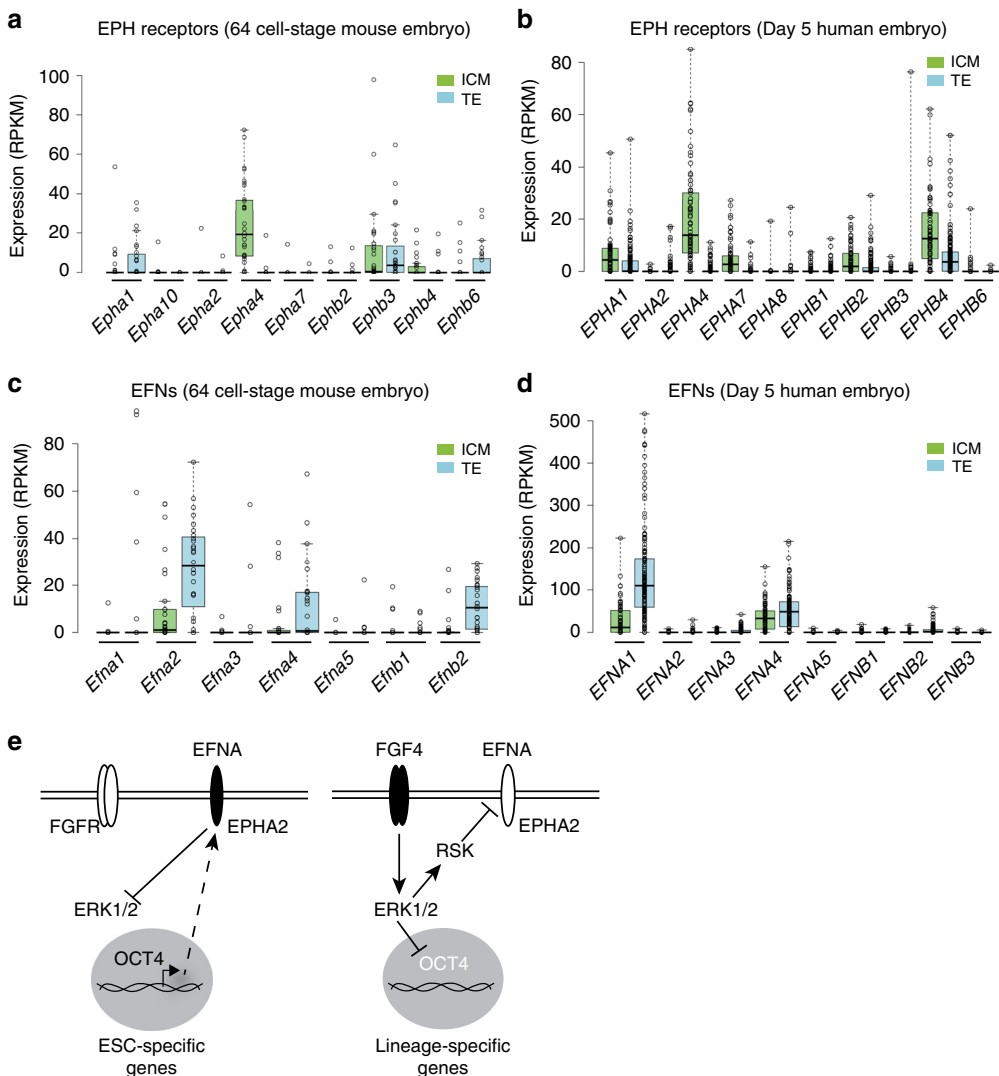

**Fig. 6 Reciprocal expression of EPH receptors and ligands in pluripotent and lineage- specified cells of early mouse and human embryos.** Expression (RPKM) of EPH (**a, b**) or EFN (**c, d**) mRNA in the inner cell mass (ICM) or trophectoderm (TE) of 64-cell mouse embryos (**a, c**) or 5-day human embryos (**b, d**). **a, c** $n = 33$ biologically independent cells for ICM, and $n = 28$ biologically independent cells for TE. **b, d** $n = 73$ biologically independent cells for ICM, and $n = 142$ biologically independent cells for TE. Box-and-whisker plots show median, first and third quartiles, and maximum and minimum values. **e** EPHA2 regulation and function in mESCs. In the pluripotent state, OCT4 and other pluripotency factors promote EPHA2 receptor expression, enabling activation by EFNA ligands to support pluripotency by restraining ERK1/2. During differentiation, FGF4 drives ERK1/2–RSK activity to phosphorylate and inhibit EPHA2, whilst ERK1/2 suppresses an OCT4–EPHA2 transcriptional module to disable EPHA2 receptor expression. Source data are provided as a Source Data file.

therefore unlikely to drive pluripotency at this early embryonic stage. However, *Epha4* is specifically upregulated in the inner cell mass (E3.5), whilst *Epha2* predominates in the pre-implantation epiblast (E4.5, Supplementary Fig. 6). This is consistent with the closely related gene expression profiles of mESCs in LIF/FBS and E4.5 epiblast[44]. Therefore, EPH receptor expression is established concomitant to induction of embryonic pluripotency factors, including *Nanog* and *Sox2*[43].

Our data suggest that pluripotent and differentiated cells specifically express EPH receptors and EFN ligands, respectively. We therefore investigated expression patterns of EPH and EFN family members during early embryonic lineage specification using single-cell RNA-sequencing data from mouse[45] and human[46]. As shown previously, *Epha4*/*EPHA4* is specifically enriched in the pluripotent inner cell mass of mouse (Fig. 6a) and human (Fig. 6b) embryos, when compared with the surrounding lineage-specified TE. In contrast, *Efna2*, *Efna4* and *Efnb2* are

specifically enriched in lineage-specified mouse TE (Fig. 6c) and *EFNA1* in human TE compared with pluripotent inner cell mass (Fig. 6d)[47,48]. These findings suggest that segregated expression of EPH receptors and EFN ligands in the early embryo plays a key role in establishing and maintaining pluripotent and differentiated compartments, respectively, consistent with our findings in mESCs. In summary, we propose a model for EPH receptor function and regulation in pluripotent cells, whereby FGF4–ERK1/2–RSK signalling disables EPH signalling, thereby relieving ERK1/2 inhibition to facilitate pluripotent exit and differentiation (Fig. 6e).

**Discussion**

The molecular mechanisms by which cellular signalling networks promote differentiation of pluripotent cells into specialised cell types remain poorly understood. In order to address this in a

systematic, unbiased way, we examine the global phosphoproteomic signature associated with mESCs differentiating in response to FGF4. Our findings indicate that FGF4 extensively rewires phosphorylation networks beyond previously known pathways and targets, suggesting that many hitherto unappreciated phosphorylation events may be key regulators of mESC differentiation.

Our analysis describes a critical function for the FGF4–ERK1/2 pathway in disabling the EPH receptor signalling during mESC differentiation. We demonstrate functional significance by identifying a role for EPHA2 in maintaining pluripotency gene expression, restricting cell commitment and induction of differentiation markers, although the role of EPHA2 in lineage specification remains to be determined. We also provide evidence that EPHA2 activation supports pluripotency gene expression by inhibiting ERK1/2[28,29]. EPHA2 has been shown to recruit protein tyrosine phosphatases[34,49–51], and we show that EPHA2 activation drives recruitment of the protein tyrosine phosphatase SHP2/PTPN11. As SHP2 is a key scaffold for ERK1/2 activation, EPHA2 may sequester SHP2 from key FGF-dependent phosphotyrosine sites that are required for ERK1/2 signalling. Interestingly, this contrasts with a recently described role of secreted EPHA7, which disrupts activation of ERK1/2 by EPH–EFN signalling during somatic cell reprogramming[52], highlighting the paradoxical nature of EPH–EFN signalling in distinct biological contexts.

FGF4 disrupts EPH receptor signalling via a bimodal mechanism, focussing on the key EPHA2 receptor tyrosine kinase, which we show is the critical mediator of EFN signalling responses in mESCs. FGF4–ERK1/2 activates the RSK kinase to inhibit EPHA2 activation via S/T phosphorylation of a key regulatory motif, which integrates signals from several other kinases[37]. Furthermore, we find that *Epha2* is a key target of OCT4 and other components of the pluripotency gene regulatory network in mESCs. As such, FGF4–ERK1/2 signalling downregulates *Epha2* during differentiation by suppressing pluripotency factor expression. By focussing on EPHA2, this dual-inhibitory system effectively shuts down mESC responses to EFN ligands during mESC differentiation. We therefore propose that FGF4–ERK1/2 functionally rewires EPH receptor signalling to promote mESC differentiation. Interestingly, this inhibitory role for FGF signalling appears to be specific to EPHA2 in mESCs, as FGFR–ERK1/2 signalling promotes activation of the related EPHB2 receptor in a different cellular system[53].

In summary, we report a function for EPH–EFN signalling in maintaining pluripotency. We also provide evidence that a similar mechanism operates in pluripotent cells of the early embryo, where EPH receptors and EFN ligands are specifically enriched in pluripotent and lineage-specified cells, respectively. Specifically, pluripotent inner cell mass cells express EPH receptors, whilst the surrounding TE cells express EFN ligands, suggesting that EPH–EFN signalling may safeguard pluripotency in the inner cell mass via contact with adjacent TE cells. Importantly, TE lineage specification is also dependent on activation of the ERK1/2 pathway[54,55], suggesting that the regulatory mechanism identified in mESCs also operates in early embryos. Furthermore, EFN ligand engagement by EPH receptors initiates bidirectional signalling[11] to drive segregation of EPH- and EFN-expressing populations[12]. Thus, regulation of EPH–EFN by ERK1/2 signalling may be involved in compartmentalisation of the early embryo. This integrated system would allow organisation of pluripotent and lineage-specified cells into the distinct cellular compartments that are essential for proper embryonic development[13].

## Methods

**Cell culture**. CCE mouse mESCs were maintained on gelatin-coated plates in media containing 100 ng/mL GST-tagged Leukemia Inhibitory Factor (LIF), 10%

Fetal Bovine Serum (FBS; Gibco) and 5% knockout serum replacement (Invitrogen). Embryoid bodies (EBs) were formed by aggregation of 1200 cells in 20 μl of mESC medium without LIF forming hanging drops[56]. At day 4, EBs were transferred onto gelatin-coated plates and maintained for a further 2–6 days. WT mESCs were converted from LIF/FBS to 2i culture conditions (N2B27 with 1 μM PD0325901 and 1 μM CHIR99021)[39], and 2i differentiation performed by culturing cells in N2B27 media alone. *Fgf4⁻/⁻* mESCs were differentiated in mESC medium without LIF containing 100 ng/ml FGF4 (Peprotech). Cell cultures used in this paper were tested for mycoplasma infection.

**Growth curves and MTS assay**. mESCs were plated in LIF/FBS medium at $5 \times 10^4$ cells/cm², collected by trypsinisation every day and counted using an automated counter (Invitrogen). Population-doubling time was calculated between days 1 and 2 (exponential growth) as follows: Doubling time = $T \ln2/\ln$ (Xe/Xb) where $T$ = incubation time in any unit, Xb = cell number at the beginning and Xe = cell number at the end.

MTS assay was performed in parallel in 96-well plates using CellTiter-Glo® Luminescent Cell Viability Assay as indicated by the manufacturers.

**mESC commitment assay**. The mESC commitment assay protocol was based on ref.[27]. mESCs were plated in 2i medium on gelatin-coated plates at $2 \times 10^4$ cells/cm². After 24 h, cells were washed with phosphate-buffered saline (PBS) and cultured in N2B27 media for 96 h or 2i for 72 h, respectively. In total, 10% of 2i-cultured cells or N2B27-cultured cells were replated in 2i medium and cultured for a further 72 h. Cells were then fixed in 4% PFA–PBS and stained with alkaline phosphatase (AP) detection kit (Sigma-Aldrich) according to the manufacturer's instructions. Plates were imaged, AP signal was quantified using Image Lab software (Bio-Rad) and individual colonies counted using ImageJ software (NIH). mESC commitment was estimated by plotting AP staining intensity from 2i control, and N2B27-differentiated cells replated in 2i at an equivalent density (10% of total replated in 2i medium for 3 days). Relative commitment was calculated by quantifying AP staining intensity for *Epha2⁺/⁺* and *Epha2⁻/⁻* mESCs following N2B27 differentiation, and replating of either 10% of cells in 2i for a further 72 h, or signal normalised to *Epha2⁺/⁺* mESCs in each case.

**Cell lysis and immunoprecipitation**. Cells were washed with PBS and lysed in ice-cold FLAG IP–MS lysis buffer (20 mM Tris-HCl, pH 7.4, 150 mM NaCl, 1 mM EDTA, 1% NP-40, 0.5% Na Deoxycholate, 10 mM β-Glycerophosphate, 50 mM NaF, 10 mM NaPPi, 2 mM Sodium Orthovanadate, 10% Glycerol and complete protease inhibitor tablets [Roche]). EPHA2 was immunoprecipitated for 2 h with 1.5 μg of anti-EPHA2 antibody and 10 μl of Protein G Agarose. Immunoprecipitates were washed three times with lysis buffer prior to immunoblot analysis.

**SDS-PAGE and immunoblotting**. Proteins resolved by sodium dodecyl sulfate polyacrylamide gel electrophoresis (SDS-PAGE) were transferred to polyvinylidene difluoride (PVDF) membranes at 30 V for 90 min. PVDF membranes were blocked with 3% skimmed milk, incubated with primary antibody (Supplementary Table 2) at 4 °C overnight and washed three times in tris-buffered saline with 0.1% (v/v) Tween-20 (TBST). Membranes were incubated in the appropriate secondary antibody, washed three times in TBST and processed using Immobilon for enhanced chemiluminescence detection (Millipore). Immunoblots were imaged using the Chemidoc gel imaging system (Biorad).

**Immunofluorescence**. Cells were seeded on gelatin-coated coverslips, and fixed with PBS 4% PFA [w/v], permeabilised in PBS 0.5% Triton X-100 [v/v] for 5 min at room temperature, blocked with 3% bovine serum albumin [w/v] in PBS and incubated with EPHA2 antibody (R&D Systems) at 1:200 in blocking buffer for 2 h at room temperature. Donkey anti-goat Alexa-488 (Thermo Fisher Scientific) was used as a secondary antibody at 1:500 in blocking buffer for 1 h. DAPI at 1:10,000 was used for nuclear staining. Cells were mounted using FluorSave reagent (Millipore), images acquired by a Zeiss 710 confocal microscope and images processed using ImageJ.

**RNA extraction and qRT-PCR**. RNA was extracted using the OMEGA total RNA kit and reverse transcribed using iScript reverse transcriptase (Biorad) according to the manufacturer's instructions. qPCR was then performed using SsoFast™ Eva-Green® Supermix (Bio-Rad). See Supplementary Table 3 for a list of primers. ΔCt values using GAPDH as a reference gene were used to analyse relative expression, and the 2 − ΔΔCt (Livak) method used to normalise to control when required.

**Phosphoproteomic profiling**. *Fgf4⁻/⁻* mESCs were cultured in LIF/FBS, and then starved for 4 h before stimulation with 100 ng/ml FGF4 (Peprotech). Cells were lysed, proteins extracted and trypsin digested. Phosphopeptide enrichment was done using TiO₂ microspheres (GL Sciences) followed by TMT labelling (Thermo Fisher Scientific). Samples were fractionated using basic C18 reverse-phase (bRP) chromatography and subjected to LC–MS/MS. Acquired LC–MS data were analysed using Proteome Discoverer software v2.2 (Thermo Fisher Scientific). Identification of significantly modified peptides after 5 and 20 min of stimulation was

done using the limma package[57] (Bioconductor). See Supplementary methods for further detail.

**Quantitative total cell proteomics**. Quantitative total cell proteomics analysis was conducted as described[19]. mESCs were cultured in LIF/FBS, lysed, boiled and sonicated before alkylation with iodoacetamide. Lysates were subjected to the SP3 procedure for protein clean-up before elution digest with LysC and Trypsin. TMT labelling and peptide clean-up were performed according to the SP3 protocol. TMT samples were fractionated using offline high-pH reverse-phase chromatography. Peptides were separated into fractions and analysed by LC–MS. The data were processed, searched and quantified with the MaxQuant software package, version 1.5.3.30. Proteins and peptides were identified using the UniProt *mouse* reference proteome database (SwissProt and Trembl accessed on 24.03.2016), and the contaminants database integrated in MaxQuant using the Andromeda search engine. The false-discovery rate was set to 1% for positive identification of proteins and peptides with the help of the reversed mouse Uniprot database in a decoy approach. Copy numbers of EPH/EFN proteins were then calculated. See Supplementary methods procedures for further detail.

**EFN interaction proteomics**. Recombinant EFNA1 and/or EFNB1 ligands (R&D systems) were clustered using human Fc IgG (Jackson ImmunoResearch) on ice for 1 h. mESCs cultured under standard conditions were lysed in FLAG IP–MS lysis buffer; clustered EFN ligands added along with Protein G Agarose and incubated for 1 h at 4 °C. Beads were collected by centrifugation at 1000g for 3 min, supernatants removed and beads washed three times in lysis buffer. Samples were analysed by SDS-PAGE, and gels were either transferred for immunoblotting, or stained with Coomassie blue, and gel slices in the 70–130-kDa range were excised and prepared for LC–MS/MS analysis.

**Preparation of SDS-PAGE LC–MS/MS samples**. Protein samples were resolved by SDS-PAGE and stained with Coomassie R-250 (Novex). Protein bands for LC–MS/MS analysis were cut into ~1-mm cubes, and washed successively in water, 50% acetonitrile, 0.1 M NH$_4$HCO$_3$ and 50% acetonitrile/50 mM NH$_4$HCO$_3$. All washes are 0.5 mL for 10 min per gel band; remove all of the liquid between washes. Protein gel pieces were incubated in 10 mM DTT/0.1 M NH$_4$HCO$_3$ at 37 °C for 20 min, then proteins were alkylated in 50 mM iodoacetamide, 0.1 M NH$_4$HCO$_3$, for 20 min at room temperature. Gel pieces were washed in 50 mM NH$_4$HCO$_3$ and then 50 mM NH$_4$HCO$_3$/50% acetonitrile. Once colourless, gel pieces were shrunk in 0.3 mL of acetonitrile for 15 min, dried in a Speed-Vac and swelled in 25 mM triethylammonium bicarbonate containing 5 μg/mL of Trypsin and incubated at 30 °C overnight. An equivalent volume of acetonitrile was added, the supernatant removed, frozen at −80 °C and dried in a Speed-Vac till dry. Samples were stored at −80 °C prior to LC–MS/MS analysis.

**siRNA knockdown**. mESCs were transfected with 50 nM siRNA (Dharmacon) using RNAiMAX (Life Technologies) according to the manufacturer's instructions, and incubated for 48 or 72 h before further analysis.

***Epha2* CRISPR/Cas9 gene targeting**. *Epha2*−/− mESCs were generated using dual-mouse *Epha2* single-guide RNAs (sgRNAs) cloned into pKN7 and pX335, respectively (Addgene; MRC Reagents & Services DU52038, DU52047). Constructs were transfected into mESCs using Lipofectamine LTX (Life Technologies), selected using 1 μg/ml puromycin for 24 h and cultured for a further 48 h. mESCs were then plated at clonal density and isolated after 8–10 days of culture. EPHA2 expression was analysed by immunoblotting, and putative positive clones were confirmed by PCR cloning and sequencing of the targeted region from genomic DNA (Supplementary Figure 7).

A CRISPR/Cas9 KI strategy was used to introduce 5-point mutations (S893/A, S898/A, T899/A, S900/A and S902/A) in the EPHA2 C-terminal region. *Epha2* sense and antisense sgRNAs were cloned into pBabeD pU6 and pX335, respectively (Addgene; MRC Reagents & Services DU60602, DU60605). Donor vectors containing *Epha2* cDNA for exons 16 and 17, either WT or 5A, followed by an IRES2 and EGFP, were cloned into pMA (Addgene; MRC Reagents & Services DU60674, DU60744) and transfected in conjunction with the mentioned sgRNA-expressing vectors using Lipofectamine LTX (Life Technologies). Cells were selected using 1 μg/ml puromycin for 24 h, and cultured for a further 48 h. Cells were analysed by FACS, and single EGFP-positive cells plated on 96-well plates. Clones were screened for EGFP, EPHA2 and EPHA2 pS898 by immunoblotting. Correct gene targeting was confirmed by genomic PCR and DNA sequencing (Supplementary Fig. 8).

**Generation of stable mESC lines**. Stable mESC lines were generated by preparing Sal1-linearised pCAGGS vectors, of which 20 μg were electroporated into mESCs. Electroporated mESCs were plated in 10-cm dishes and selected with 1 μg/ml puromycin for 24 h. Cells either underwent single-cell sorting and expansion, or were cultured at clonal density. Single-cell clones were isolated after 8–10 days in culture before further analysis.

**Statistics and reproducibility**. Graphical data were generated and analysed by GraphPad PRISM software. Continuous variables were expressed as the mean ± SEM or mean ± SD as indicated in the figure legends. Statistical analyses were performed as stated in the figure legends. P values of <0.05 were considered as statistically significant. Unless otherwise stated, representative experiments were performed at least three times with similar results. The original unprocessed and uncropped gels/blots with molecular weight marker information are provided in Supplementary Fig. 9 and as a Source Data file.

**Materials**. Many reagents generated for this study are available to request through the MRC-PPU reagents website (https://mrcppureagents.dundee.ac.uk/).

**Reporting summary**. Further information on research design is available in the Nature Research Reporting Summary linked to this article.

## Data availability
Phosphoproteomic profiling data from Fig. 1 have been deposited to the ProteomeXchange Consortium via the PRIDE partner repository with the dataset identifier PXD012069. The source data underlying all experimental results (Figs. 1c–f; 2a, c–h; 3; 4b–d, f–j; 5a, b, d, e; 6a–d and Supplementary Figs. 1a, 2, 3a–d, f–h, 4, 5a–d) are provided as a Source Data file. The authors declare that all other data supporting the findings of this study are available within the paper and its Supplementary information files.

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

## Acknowledgements

The authors would like to thank Dr. Rachel Toth and Tom Macartney (MRC-PPU Reagents and Services) for cloning, Fiona Brown (MRC-PPU Reagents and Services) for antibody production and Dr. David Campbell and Joby Vhargese (MRC-PPU) for mass-spectrometry analysis. We would also like to thank Dr. Janet Rossant (SickKids, Toronto), the late Dr. Tony Pawson (LTRI, Toronto), Prof. Vicky Cowling and Dr. Marios Stavridis (University of Dundee) for helpful discussions. R.F.-A., F.B. and G.M.F. were supported by an MRC New Investigator Research Grant MR/N000609/1 and Wellcome Trust/Royal Society Sir Henry Dale Fellowship 211209/Z/18/Z awarded to G.M.F.

## Author contributions

R.F.-A., G.M.F., F.B., P.K., M.B., H.Z., A.H. and J.H. performed experiments, analysed the data and prepared figures. E.P. analysed the data and prepared figures. A.I.L. and F.L. provided reagents, technical expertise and conceptual input. R.F.-A. and G.M.F. wrote the paper.

## Competing interests

The authors declare no competing interests.
