## [Peer Review File · Nature Communications]

Reviewers' comments:

Reviewer #1 (Remarks to the Author):

The manuscript by Findlay and coworkers describes a large-scale phosphoproteomics analysis of Fgf4 stimulated mouse ESC cultures using chemical dimethyl labelling-based quantification. Cells were stimulated for 0, 5 and 20 min and phosphopeptides enriched by TiO₂ prior to LC-MS/MS. This resulted in the identification of more than ten thousand phosphosites of which 300 were regulated more than two-fold. The major findings of the study is the identification of a bimodal EPHA2 receptor switch that promotes ESC differentiation. The authors find that FGF4-dependent ERK1/2 activation of RSK promotes inhibitory EPHA2 phosphorylation, which suppresses EPHA2 during ESC differentiation. However, FGF-dependent activation of ERK/RSK and RSK2-dependent phosphorylation of EphA2 on S897 is well established in the literature (see for example, Zhou Y, et al. (2015) Crucial roles of RSK in cell motility by catalyzing serine phosphorylation of EphA2. *Nat Commun* 6, 7679), although not in the context of ES cell differentiation it still takes away much of the novelty. Importantly, the authors need to do a better job in describing and presenting the quantitative phosphoproteomics screen, which is the foundation of all subsequent analyses presented in the manuscript – see major points below. Consequently, I cannot recommend publication of the manuscript in its current form as the phosphoproteomics dataset simply does not keep up to the community standards.

MAJOR POINTS:

Firstly, although the phosphoproteome analysis is based on heavy isotope-tagged dimethyl labelling quantification no biological replicates for any of the time-points are presented, which is a bit disturbing. N=1 experiments are not acceptable in state-of-the-art phosphoproteomics, where a minimum of three biological replicates per condition is a minimum to demonstrate that the claims made based on the changing phosphosites can be reproduced in different mESC cultures on different days. The lack of replicates in this otherwise interesting phosphoproteomics paper is a major weakness, which needs to be addressed before considering publication in any journal.

Secondly, lack of availability of raw MS data. No raw MS data/output files are provided, which makes it impossible for other researchers to evaluate and scrutinize the quality of the data. The raw LC-MS/MS data files and the associated search files and output tables should be made available to the proteomics community, for example, through the MASSIVE or ProteomeXchange repositories.

Thirdly, in addition to making the raw MS data and results available, the material and methods around parameters of MS analysis is also incomplete as it is completely unclear what fractionation method was used and which MS settings employed. This needs to be addressed.

Finally, the presented study is not the first one to investigate ES cell differentiation using quantitative phosphoproteomics. Overlap of regulated phosphorylation sites with previous large-scale phosphoproteomics datasets on ES cell differentiation using Fgf-stimulation (Ding VM et al, *PLoS ONE* 2011; Zoumaro-Djayoon AD et al, *Proteomics* 2011) needs to be presented.

Reviewer #2 (Remarks to the Author):

The authors have identified a bimodal mechanism by which FGF4 signalling inhibits EphA2 signalling in embryonic stem cells cultures to allow the transition from pluripotency to differentiation. The phosphoproteomics screen that lead to this discovery was well designed and performed. In general,

the quality of the data is very high and the results were presented clearly. The major conclusions that EphA2 is the major trans-membrane kinase in cultured ESCs and functions to support pluripotency gene expression and restrain lineage commitment in this culture system, are well supported by the data.

However, the authors have not added sufficient experimental data to support the conclusion that “a similar mechanism might operate in pluripotent cells of the early embryo” (page 12).

I have the following major points:

1. Under standard mESC conditions, EphA2 seems to be active, since results in this manuscript indicate that differentiating EphA2^{-/-} mESCs show altered expression of pluripotency genes such as KLF4. The authors fail to demonstrate and discuss how EphA2 signaling is activated under standard mESC conditions. Do mESCs express ephrinAs (as the model in Figure 6 implies)? Would disruption of ephrinAs expression cause similar alterations as disruption of EphA2 expression? Do ephrinAs activate EphA2 in trans (ephrinAs and EphA2 on opposing cells) or in cis (ephrinAs and EphA2 on the same plasma membrane)?

2. These above points are important, as Eph expression seems very dynamic in the early embryo and as commonly observed interactions with ephrins lead to Eph internalization, recycling, and/or degradation. Unlike standard mESC conditions, the authors find that EphA2 expression in E2.5 morula is undetectable, despite the fact that morula cells are pluripotent. How does that result fit into the model? Which mechanism maintains pluripotency in morula cells in the absence of Eph expression? What mechanism downregulates EphA2 expression as ESCs develop into a morula? Likewise, in E3.5 inner mass cells (also pluripotent and expressing FGF4), other Eph genes are abundantly expressed whereas EphA2 is not (Fig.5). EphB3 is particularly abundant and should respond to ephrinBs, not ephrinAs. I am confused about how this expression pattern fits the idea of EphA2 being the major kinase supporting pluripotency in the early embryo.

3. EphA2 knockout mice are viable. This argues that either EphA2 is dispensable for supporting pluripotency, or EphA2 is functionally redundant with other Eph receptors. Regardless, this limits the relevance of the work significantly.

4. The authors show that multiple serine phosphorylations of EphA2 induced by FGFR can block the activation of EphA2 based on EphA2 autophosphorylation (Fig. 4). This important result needs further investigation. The authors used soluble preclustered ephrinA1 to stimulate EphA2. Is this the physiological way to activate EphA2? Alternatively, is endogenous ephrinA1 tethered to the surface of nearby cells? I would like to see the effects of EphA2 serine phosphorylation on EphA2 activation in a cell-cell stimulation assay in which ephrinA-expressing cells are mixed with EphA2-expressing cells.

Reviewer #3 (Remarks to the Author):

Fernandez-Alonso et al. report Epha2 as a critical phosphorylation target of Fgf/ERK signalling in mouse ES cells and suggest a function of Ephrin signalling to counteract Fgf/ERK mediated induction of mouse ES cell differentiation. Epha2 was found to be essential for Ephrin signalling in mESCs. Although apparently without phenotypic effect in ESC self-renewal, loss of Epha2 results in aggravated Klf4 and Dnmt3a downregulation during differentiation, suggesting a role of Ephrin signalling in supporting naïve pluripotency. Further the authors show data indicative of a role for Oct4 in driving Epha2 expression. In summary, this suggests Epha2 as a highly interconnected hub (regulated by

Fgf/ERK and Oct4 - regulating Fgf/ERK and the naïve pluripotency TF-circuit) to mediate the switch from pluripotency to differentiation.

In summary, the results presented are highly interesting and data appears to be of high technical quality. If, as the authors suggest, Ephrin signalling is regulated by Fgf/ERK, negatively regulates Fgf/ERK and is sustains naïve pluripotency, that would certainly make this manuscript interesting for a wide audience.

However, based on the data presented it is not possible to assess the real impact of Ephrin signalling on both pluripotency and/or differentiation. Ephrin signalling appears to be somehow linked to Fgf signalling and the transition from naïve to primed pluripotency, but the extent to which Epha2 is required and / or sufficient to drive cell fate change remains unclear. Further, the mechanism of action of how Ephrin signalling interacts with the ERK signalling cascade remains elusive. The analysis performed is rather superficial and does not provide substantial mechanistic insight into the function of Epha2 in ES cells.

Major points:

1) The effect of disruption of Ephrin signalling on the exit and maintenance of pluripotency are not sufficiently worked out. More assays, e.g. including commitment assays (Betschinger et al., 2013) and immunofluorescence based assays measuring expression kinetics of several naïve pluripotency markers during differentiation should be performed. RNASeq experiments should be performed to assess the impact of loss of Ephrin signalling on self-renewal and differentiation.

Figure 3CA-C show the effect of loss of Epha2 on the expression kinetics of naïve and primed pluripotency markers. These should include a more comprehensive analysis of a set of naïve pluripotency markers using qPCR. In addition, to address how Ephrin signalling integrates in the transition between primed and naïve pluripotency transcriptionally, it will be important to concomitantly assess the expression kinetics of Ephrin signalling components.

2) From the presented data, the actual impact of Epha2 on stabilising self renewal remains elusive. A central question is whether, similar to pStat3 and pAkt, Fgf induced inactivation resistant Epha2 (S5A?) is sufficient to drive ES cell self-renewal in the absence of LIF.

3) The authors claim that Ephrin inactivation is induced by Fgf/ERK. Data showing that Epha2 is indeed an Fgf/ERK target is clear. However, whether that regulation is of functional consequence is not. Also, EPHA2 is proposed to promote mESC pluripotency by specifically inhibiting ERK1/2 signalling. I can see no evidence for any specificity in Ephrin function in that respect. Figure 3D does indeed show that induction of Ephrin signalling reduces ERK phosphorylation (but not pStat3), but this is not evidence for a specific interaction.

If the proposed interactions are functional, then it could be predicted that addition of exogenous Fgf in experiments performed in Fig3A should phenocopy loss of Epha2 in WT cells. Also, in case Ephrin signaling functions in restricting Fgf activity, the increase in Dnmt3a upregulation and Klf4 downregulation kinetics should be reversed by addition of a MEK inhibitor in Epha2 KO cells. Further, genetic co-depletion experiments (Fgf4:Epha2 dKO) could be performed to show an actual genetic interaction of Fgf and Ephrin signalling.

4) The claim that Ephrin signalling functions in restricting epiblast and mesendoderm differentiation, based on expression levels of two markers, lacks substance. In a developmental trajectory, ES cells will firstly upregulate postimplantation epiblast markers (e.g. Fgf5) and then induce mesendodermal

genes (e.g. Brachyury). However, as Epha2 ES cells have a general differentiation defect at the exit from pluripotency, it is not clear, whether there is any additional role of Ephrin signalling at the time where differences in T and Fgf5 expression are observed. In case a role for Epha2 in germ layer specification needs to be addressed, proper analyses have to be performed. The first step would be to use a much wider panel of marker genes for qPCR, including early endodermal markers and further epiblast and mesoderm markers. To assess this experiment properly, also expression kinetics of pluripotency markers (and Ephrin signalling components) need to be recorded.

5) A causal link between Oct4 and Epha2 expression cannot be constructed from siRNA experiment (Fig5), as acute loss of Oct4 has a massive impact on cell identity and induces differentiation. In case solely the presence of Oct4 was driving Epha2 expression, then transcription would not be downregulated 2 days after 2i release as shown in Fig 4E, as at that time Oct4 expression levels remain at 2i levels.

Of note, the two Sox2siRNAs behave very differently. Is this consistent between assays? If so another siRNA should be used.

As a matter of fact, I doubt that experiments presented in Figure5 contribute any substantial information at all to the manuscript and I suggest to focus on the main message of the paper and delete this part.

6) On a technical note, S5E Ephrin TGs in FigS4B are very lowly expressed (there also seems to be a double band, where all other bands show only one band), therefore data are hard to interpret.

7) Growth curves should be performed for all analysed ES cells to exclude an impact of proliferation on the observed phenotypes.

8) What do the authors mean with compartmentalisation of cells with distinct developmental destinies? To me this makes no sense. Where do they show compartmentalisation as compared to whole population responses in the manuscript? This sentence should be removed from the introduction. A potential, speculative, role in compartmentalisation is mentioned in the discussion anyways.

Minor points:

1) Fig1E: The y-axis is labelled relative abundance, but relative to what?

2) Fig1F: There are two sets of bars for Epha2; why is that?

3) Fig3D: the reference bar does not have an error bar, which should be included (this is also the case for several other plots).

4) Sox2 siRNA results are not in line with published data that show that loss of Sox2 leads to loss of Oct4 expression (Masui et al., 2007). This should be addressed.

5) Significance of Fgf5 and Brachyury upregulation is indicated (Fig3C), but is significance based on comparison with WT or with rescue cell lines?

6) Why is the duration of the timecourse different between the Western and IP-Western panels in Fig 3D. Is this a different experiment?

7) Fgfr1 has recently been reported to be the main driver of Fgf/ERK signalling in ES Cells (Molotov et al., 2017); therefore it is rather surprising that the authors detect an average of only < 5 molecules (maybe even less, this is hard to tell from the graph) per cell in Fig2a. Can they comment on that?

8) Fig2D appears to show an unspecific band, which should be indicated as such.

Reviewer #1 (Remarks to the Author):

The manuscript by Findlay and coworkers describes a large-scale phosphoproteomics analysis of Fgf4 stimulated mouse ESC cultures using chemical dimethyl labelling-based quantification. Cells were stimulated for 0, 5 and 20 min and phosphopeptides enriched by TiO₂ prior to LC-MS/MS. This resulted in the identification of more than ten thousand phosphosites of which 300 were regulated more than two-fold. The major findings of the study is the identification of a bimodal EPHA2 receptor switch that promotes ESC differentiation. The authors find that FGF4-dependent ERK1/2 activation of RSK promotes inhibitory EPHA2 phosphorylation, which suppresses EPHA2 during ESC differentiation. However, FGF-dependent activation of ERK/RSK and RSK2-dependent phosphorylation of EphA2 on S897 is well established in the literature (see for example, Zhou Y, et al. (2015) Crucial roles of RSK in cell motility by catalyzing serine phosphorylation of EphA2. Nat Commun 6, 7679), although not in the context of ES cell differentiation it still takes away much of the novelty. Importantly, the authors need to do a better job in describing and presenting the quantitative phosphoproteomics screen, which is the foundation of all subsequent analyses presented in the manuscript – see major points below. Consequently, I cannot recommend publication of the manuscript in its current form as the phosphoproteomics dataset simply does not keep up to the community standards.

MAJOR POINTS:

Firstly, although the phosphoproteome analysis is based on heavy isotope-tagged dimethyl labelling quantification no biological replicates for any of the time-points are presented, which is a bit disturbing. N=1 experiments are not acceptable in state-of-the-art phosphoproteomics, where a minimum of three biological replicates per condition is a minimum to demonstrate that the claims made based on the changing phosphosites can be reproduced in different mESC cultures on different days. The lack of replicates in this otherwise interesting phosphoproteomics paper is a major weakness, which needs to be addressed before considering publication in any journal.

We thank the referee for their constructive comments on our paper. We have now performed a state-of-the-art phosphoproteomic analysis in biological triplicate using TMT 9-plex isobaric labeling (Figure 1 and Supplementary Figure 1). We believe that this represents a robust and valuable resource for the proteomics and stem cell communities

Secondly, lack of availability of raw MS data. No raw MS data/output files are provided, which makes it impossible for other researchers to evaluate and scrutinize the quality of the data. The raw LC-MS/MS data files and the associated search files and output tables should be made available to the proteomics community, for example, through the MASSIVE or ProteomeXchange repositories.

We will upload all raw MS data to the Proteome Exchange Consortium via the PRIDE Partner repository. This is currently in progress.

Thirdly, in addition to making the raw MS data and results available, the material and methods around parameters of MS analysis is also incomplete as it is completely unclear what fractionation method was used and which MS settings employed. This needs to be addressed.

We have now provided comprehensive mass-spectrometry parameters and methods in the Appendices.

Finally, the presented study is not the first one to investigate ES cell differentiation using quantitative phosphoproteomics. Overlap of regulated phosphorylation sites with previous

large-scale phosphoproteomics datasets on ES cell differentiation using Fgf-stimulation (Ding VM et al, PLoS ONE 2011; Zoumaro-Djayoon AD et al, Proteomics 2011) needs to be presented.

We thank the referee for this useful suggestion, and have now provided Venn diagrams demonstrating the degree of overlap between our dataset and those of Ding *et al.* and Zoumaro-Djayoon *et al.* (Supplementary Figure 1B). This illustrates that although there is some overlap with these datasets, our more in-depth analysis has the potential to provide substantial new insights into functions of FGF signaling in ES cells.

Reviewer #2 (Remarks to the Author):

The authors have identified a bimodal mechanism by which FGF4 signalling inhibits EphA2 signalling in embryonic stem cells cultures to allow the transition from pluripotency to differentiation. The phosphoproteomics screen that led to this discovery was well designed and performed. In general, the quality of the data is very high and the results were presented clearly. The major conclusions that EphA2 is the major trans-membrane kinase in cultured ESCs and functions to support pluripotency gene expression and restrain lineage commitment in this culture system, are well supported by the data.

However, the authors have not added sufficient experimental data to support the conclusion that “a similar mechanism might operate in pluripotent cells of the early embryo” (page 12). We thank the referee for acknowledging the high quality of our data supporting the role of EPHA2 in ESCs. We agree that further investigating the function of EPH-EFNA signaling in the early embryo is an important question to be addressed, and now provide exciting new data concerning the function of the EPH-EFN system *in vivo* (Figure 6).

I have the following major points:

1. Under standard mESC conditions, EphA2 seems to be active, since results in this manuscript indicate that differentiating EphA2^{-/-} mESCs show altered expression of pluripotency genes such as KLF4. The authors fail to demonstrate and discuss how EphA2 signaling is activated under standard mESC conditions. Do mESCs express ephrinAs (as the model in Figure 6 implies)? Would disruption of ephrinAs expression cause similar alterations as disruption of EphA2 expression? Do ephrinAs activate EphA2 in trans (ephrinAs and EphA2 on opposing cells) or in cis (ephrinAs and EphA2 on the same plasma membrane)?

We thank the reviewer for raising the key issue of how EPHA2 is activated in mESCs. We now provide compelling new evidence in Figure 2g that EPHA2 is tyrosine phosphorylated and active under standard mESC culture conditions. This is likely mediated by A-type Ephrin ligands, as new data presented in Figure 4h shows that several Ephrin family members are expressed at low levels in mESCs, whilst over-expression of EFNA1 in mESCs drives EPHA2 tyrosine phosphorylation (Figure 3c). We also show that activation of EPHA2 by EFNA1 is functionally relevant, promoting expression of pluripotency genes NANOG and KLF4 (Figure 3c), and naive mESC morphology with strong alkaline phosphatase activity in commitment assays (Supplementary Figure 3D). Finally, EPHA2 activation by EFNA1 is likely to occur in trans, as cis interactions between ligand and receptor are associated with signal attenuation (Yin, Y *et al.*, 2004; Hornberger MR *et al.*, 1999). Indeed, we provide new data to this effect, as EFNA1-expressing *Epha2*^{-/-} mESCs cells drive EPHA2 tyrosine phosphorylation when mixed with wild-type cells (Figure 2h).

2. These above points are important, as Eph expression seems very dynamic in the early embryo and as commonly observed interactions with ephrins lead to Eph internalization,

recycling, and/or degradation. Unlike standard mESC conditions, the authors find that EphA2 expression in E2.5 morula is undetectable, despite the fact that morula cells are pluripotent. How does that result fit into the model? Which mechanism maintains pluripotency in morula cells in the absence of Eph expression? What mechanism downregulates EphA2 expression as ESCs develop into a morula? Likewise, in E3.5 inner mass cells (also pluripotent and expressing FGF4), other Eph genes are abundantly expressed whereas EphA2 is not (Fig.5). EphB3 is particularly abundant and should respond to ephrinBs, not ephrinAs. I am confused about how this expression pattern fits the idea of EphA2 being the major kinase supporting pluripotency in the early embryo.

Our data indicating that EPHA2 is the major EPH receptor in mESCs cultured in LIF/FBS reflects the fact these cells that provide a developmental snapshot of E4.5 epiblast, rather than E2.5 morula or E3.5 inner cell mass (Boroviak *et al.*, 2014). However, we agree with the referee that EPH receptor expression in the early embryo is very dynamic and complex, and that other family members, particularly EPHA4, may also contribute to pluripotency maintenance. In support of this notion, we provide exciting new data from early mouse and human embryos, indicating that EPHA4 is enriched in the pluripotent inner cell mass, whilst EFNA-type ligands are enriched in the surrounding lineage-specified trophectoderm (Figure 6). We cannot rule out a role for EPHB3 or B-type EPH receptors in the early embryo.

Regarding the morula, these cells are pluripotent, but the mechanisms conferring pluripotency at E2.5 are poorly understood and distinct from E3.5 inner cell mass and E4.5 pre-implantation epiblast (Boroviak *et al.*, 2014). We have added a section in the text explaining that as EPH receptors are not expressed in the E2.5 morula, EPH signaling is unlikely to promote pluripotency at this early embryonic stage.

3. EphA2 knockout mice are viable. This argues that either EphA2 is dispensable for supporting pluripotency, or EphA2 is functionally redundant with other Eph receptors. Regardless, this limits the relevance of the work significantly.

Epha2^{-/-} mice display a neural phenotype, indicating that EPHA2 has a key developmental function at a later stage of development (Naruse-Nakajima *et al.*, 2001). However, compound *Eph* gene knockouts are frequently required to observe phenotypes *in vivo* (Fagotto *et al.*, 2014), suggesting redundancy or compensation. Indeed, we provide new data showing compensatory upregulation of *Epha1* in *Epha2*^{-/-} mESCs (Supplementary Figure 2C), indicating that EPHA1 or other family members may compensate for EPHA2 during early embryonic development.

4. The authors show that multiple serine phosphorylations of EphA2 induced by FGFR can block the activation of EphA2 based on EphA2 autophosphorylation (Fig. 4). This important result needs further investigation. The authors used soluble preclustered ephrinA1 to stimulate EphA2. Is this the physiological way to activate EphA2? Alternatively, is endogenous ephrinA1 tethered to the surface of nearby cells? I would like to see the effects of EphA2 serine phosphorylation on EphA2 activation in a cell-cell stimulation assay in which ephrinA-expressing cells are mixed with EphA2-expressing cells.

We agree with the referee that using soluble, preclustered EFNA1 is not a physiological way to activate EPHA2. We now provide new data showing that EFNA1 expression in mESCs (Figure 2g) and cell-cell stimulation of mESCs with *Epha2*^{-/-} mESCs expressing EFNA1 on the surface activate EPHA2 (Figure 2h). We also employ CRISPR/Cas9 technology to generate an endogenous EPHA2 knock-in mutant that cannot be phosphorylated at the FGF4-dependent S898 motif (5A). Excitingly, 5A-KI mESCs show elevated EPHA2 tyrosine phosphorylation in a cell-cell stimulation assay (Figure 4d), confirming that Ser/Thr phosphorylation at the S898 motif disrupts physiological EPHA2 activation.

Reviewer #3 (Remarks to the Author):

Fernandez-Alonso et al. report Epha2 as a critical phosphorylation target of Fgf/ERK signalling in mouse ES cells and suggest a function of Ephrin signalling to counteract Fgf/ERK mediated induction of mouse ES cell differentiation. Epha2 was found to be essential for Ephrin signalling in mESCs. Although apparently without phenotypic effect in ESC self-renewal, loss of Epha2 results in aggravated Klf4 and Dnmt3a downregulation during differentiation, suggesting a role of Ephrin signalling in supporting naïve pluripotency. Further the authors show data indicative of a role for Oct4 in driving Epha2 expression. In summary, this suggests Epha2 as a highly interconnected hub (regulated by Fgf/ERK and Oct4 - regulating Fgf/ERK and the naïve pluripotency TF-circuit) to mediate the switch from pluripotency to differentiation.

In summary, the results presented are highly interesting and data appears to be of high technical quality. If, as the authors suggest, Ephrin signalling is regulated by Fgf/ERK, negatively regulates Fgf/ERK and is sustains naïve pluripotency, that would certainly make this manuscript interesting for a wide audience.

However, based on the data presented it is not possible to assess the real impact of Ephrin signalling on both pluripotency and/or differentiation. Ephrin signalling appears to be somehow linked to Fgf signalling and the transition from naïve to primed pluripotency, but the extent to which Epha2 is required and / or sufficient to drive cell fate change remains unclear. Further, the mechanism of action of how Ephrin signalling interacts with the ERK signalling cascade remains elusive. The analysis performed is rather superficial and does not provide substantial mechanistic insight into the function of Epha2 in ES cells.

We thank the referee for acknowledging the quality of our data and their interest in the role of EPHA2 in ESCs. We agree that further investigating the function of EPH-EFN signaling in pluripotency, differentiation and regulation of ERK1/2 signalling are important questions to be addressed. We now provide exciting new data demonstrating the key role of EPH signaling in ESC commitment and stabilizing pluripotency, and show that EPHA2 intersects with the ERK1/2 pathway by recruiting the SHP2/PTPN11 protein tyrosine phosphatase.

Major points:

1) The effect of disruption of Ephrin signalling on the exit and maintenance of pluripotency are not sufficiently worked out. More assays, e.g. including commitment assays (Betschinger et al., 2013) and immunofluorescence based assays measuring expression kinetics of several naïve pluripotency markers during differentiation should be performed. RNASeq experiments should be performed to assess the impact of loss of Ephrin signalling on self-renewal and differentiation.

We thank the referee for their constructive questions regarding the wider functions of EPH-EFN signalling in mESC pluripotency and commitment. We provide new evidence for a key role of EPHA2 in restricting mESC commitment (Figure 3b), using a similar commitment assay as Betschinger *et al.* (2013). Excitingly, we also show that EPHA2 activation by EFNA1 drives expression of naïve pluripotency genes including NANOG and KLF4 (Figure 3c), and maintains naïve pluripotent cell morphology with strong Alkaline Phosphatase staining following commitment assay (Supplementary Figure S3D). Together, these experiments provide compelling evidence that EPHA2 signalling reinforces pluripotency gene expression, thereby stabilising pluripotency and delaying the onset of differentiation.

Figure 3CA-C show the effect of loss of Epha2 on the expression kinetics of naïve and primed pluripotency markers. These should include a more comprehensive analysis of a set

of naïve pluripotency markers using qPCR. In addition, to address how Ephrin signalling integrates in the transition between primed and naïve pluripotency transcriptionally, it will be important to concomitantly assess the expression kinetics of Ephrin signalling components. We now provide compelling phenotypic and gene expression data confirming the function of EPHA2 in pluripotency maintenance (Figures 3b, 3c and Supplementary 3d). We have also addressed the expression kinetics of EPH-EFN signaling components (Figure 4g and 4h) and pluripotency markers Klf4, Oct4 and Nanog (Supplementary Figure 3e) during EB differentiation. Epha2 suppression temporally correlates with loss of pluripotency marker expression, consistent with a role of Epha2 in pluripotency.

2) From the presented data, the actual impact of Epha2 on stabilising self renewal remains elusive. A central question is whether, similar to pStat3 and pAkt, Fgf induced inactivation resistant Epha2 (S5A?) is sufficient to drive ES cell self-renewal in the absence of LIF.

We provide new data showing that EPHA2 activation is sufficient to maintain naïve pluripotent cell morphology and alkaline phosphatase staining even in differentiation inducing conditions (ESC commitment assay; Figure 3c and Supplementary Figure 3D).

Furthermore, we employ CRISPR/Cas9 technology to generate an endogenous EPHA2 knock-in mutant that cannot be phosphorylated at the FGF4-dependent S898 motif (5A). Excitingly, 5A-KI mESCs display a naïve-like morphology (Figure 4e), and elevated expression of pluripotency markers (Fig 4f).

3) The authors claim that Ephrin inactivation is induced by Fgf/ERK. Data showing that Epha2 is indeed an Fgf/ERK target is clear. However, whether that regulation is of functional consequence is not.

As the referee points out, we clearly demonstrate that EPHA2 inhibitory phosphorylation and transcriptional suppression is regulated by FGF4-ERK1/2 signalling. Using our EPHA2 5A-KI mESCs, we now provide conclusive evidence that FGF4-ERK1/2-dependent phosphorylation of the S898 motif disrupts EPHA2 activation (Figure 4d), pluripotency marker expression (Fig 4f) and naïve-like morphology (Figure 4e). Furthermore, we provide additional evidence that EPHA2 disruption interferes with pluripotency maintenance (Figure 3c). These data confirm that FGF4-ERK1/2-dependent transcriptional and post-translational regulation of EPHA2 have major functional consequences.

Also, EPHA2 is proposed to promote mESC pluripotency by specifically inhibiting ERK1/2 signalling. I can see no evidence for any specificity in Ephrin function in that respect. Figure 3D does indeed show that induction of Ephrin signalling reduces ERK phosphorylation (but not pStat3), but this is not evidence for a specific interaction.

We provide exciting new insight into the mechanism by which EPH-EFN signaling inhibits ERK1/2. EPHA2 activation in mESCs drives recruitment of the SHP2/PTPN11 protein tyrosine phosphatase (Figure 3f), which functions as a key FGFR scaffold (Yasui, Findlay *et al*, Mol Cell 2014) to promote ERK1/2 activation and ESC differentiation (Saxton *et al*, EMBO J 1997; Qu *et al*, Oncogene 1998; Burdon *et al*, Dev Biol 1999; Chan *et al* Blood 2003). Our data therefore suggest that EPHA2 inhibits ERK1/2 activation by sequestering SHP2 away from FGFR complexes.

If the proposed interactions are functional, then it could be predicted that addition of exogenous Fgf in experiments performed in Fig3A should phenocopy loss of Epha2 in WT cells.

We have shown previously that addition of exogenous FGF4 to *Fgf4*^{-/-} mESCs (which are defective for autocrine FGF4 signalling) phenocopies *Epha2* knockout by elevating DNMT3B and suppressing KLF4 expression (Findlay *et al* 2013; Figure 1D). We have added this information and citation to the text.

Also, in case Ephrin signaling functions in restricting Fgf activity, the increase in Dnmt3a upregulation and Klf4 downregulation kinetics should be reversed by addition of a MEK inhibitor in Epha2 KO cells. Further, genetic co-depletion experiments (Fgf4:Epha2 dKO) could be performed to show an actual genetic interaction of Fgf and Ephrin signalling.

We provide new data in Supplementary Figure 3F showing that FGF4-ERK1/2 inhibition (using a MEK1/2 inhibitor) in *Epha2*^{-/-} mESCs reverses the effect on DNMT3b and KLF4, indicative of a (chemical) genetic interaction between FGF4-ERK1/2 signalling and EPHA2.

4) The claim that Ephrin signalling functions in restricting epiblast and mesendoderm differentiation, based on expression levels of two markers, lacks substance. In a developmental trajectory, ES cells will firstly upregulate postimplantation epiblast markers (e.g. *Fgf5*) and then induce mesendodermal genes (e.g. *Brachyury*). However, as Epha2 ES cells have a general differentiation defect at the exit from pluripotency, it is not clear, whether there is any additional role of Ephrin signalling at the time where differences in T and *Fgf5* expression are observed. In case a role for Epha2 in germ layer specification needs to be addressed, proper analyses have to be performed. The first step would be to use a much wider panel of marker genes for qPCR, including early endodermal markers and further epiblast and mesoderm markers. To assess this experiment properly, also expression kinetics of pluripotency markers (and Ephrin signalling components) need to be recorded.

We provide new data tracking the developmental kinetics of *Epha2*^{-/-} embryoid bodies (EBs). This shows that *Epha2*^{-/-} EBs not only upregulate post-implantation epiblast (*Fgf5*) and early mesendodermal (*Brachyury*) genes at early time points, but also display increased expression of mesendodermal (*Mixl1*) and early endodermal genes (*Cer1*) (Figure 3c). We also provide pluripotency marker kinetics (*Nanog*, *Oct4* and *Klf4*), showing that pluripotency markers are quickly extinguished during EB differentiation as expected (Supplementary Figure 3E). Therefore, EPHA2 regulates ESC differentiation kinetics in a clearly defined developmental trajectory, consistent with a general role of EPHA2 in pluripotency maintenance and differentiation. However, a later function for EPHA2 in germ layer specification cannot be ruled out, which we clearly indicate in the text.

5) A causal link between Oct4 and Epha2 expression cannot be constructed from siRNA experiment (Fig5), as acute loss of Oct4 has a massive impact on cell identity and induces differentiation. In case solely the presence of Oct4 was driving Epha2 expression, then transcription would not be downregulated 2 days after 2i release as shown in Fig 4E, as at that time Oct4 expression levels remain at 2i levels.

We agree with the referee that OCT4 has profound effects on gene expression, and that pluripotency factors other than OCT4 may be involved in Epha2 transcriptional regulation. Indeed, NANOG expression closely correlates with EPHA2 expression (Supplementary Figure 4G). Furthermore, we show ChIP-SEQ data in Figure 5c indicating that OCT4, NANOG and several other key pluripotency transcription factors engage key regulatory sequences within the *Epha2* promoter. This establishes a specific and direct molecular connection from OCT4 and other pluripotency factors to *Epha2* gene regulation.

Of note, the two Sox2siRNAs behave very differently. Is this consistent between assays? If so another siRNA should be used.

As shown in Figure 5a, the degree of Sox2 knockdown is different. However, even when Sox2 is efficiently knocked down (lane 7th in figure 5a) we do not see any effect on EPHA2 expression.

As a matter of fact, I doubt that experiments presented in Figure5 contribute any substantial

information at all to the manuscript and I suggest to focus on the main message of the paper and delete this part.

We believe that data in Figure 5 establishes a specific and direct molecular connection between core pluripotency factors and the *Epha2* gene, which is critical to understand mechanisms controlling *Epha2* expression in mESCs. However, this section can be removed to add clarity if required.

6) On a technical note, S5E Ephrin TGs in FigS4B are very lowly expressed (there also seems to be a double band, where all other bands show only one band), therefore data are hard to interpret.

EPHA2 S5E can display different mobility on SDS-PAGE due to the increased negative charge. However, we quantify both bands within the doublet and use this total signal to normalize EPHA2 tyrosine phosphorylation to expression level.

7) Growth curves should be performed for all analysed ES cells to exclude an impact of proliferation on the observed phenotypes.

We now provide growth curve data confirming that *Epha2*^{-/-} mESC lines have similar proliferative capacity as *Epha2*^{+/+} mESCs (Supplementary Figure 3B and 3C).

8) What do the authors mean with compartmentalisation of cells with distinct developmental destinies? To me this makes no sense. Where do they show compartmentalisation as compared to whole population responses in the manuscript? This sentence should be removed from the introduction. A potential, speculative, role in compartmentalisation is mentioned in the discussion anyways.

We thank the referee for highlighting this confusing sentence. We have now removed it from the introduction.

Minor points:

1) Fig1E: The y-axis is labelled relative abundance, but relative to what?

We have now performed a new phosphoproteomic analysis as requested by referee 1, and include a modified Figure 1.

2) Fig1F: There are two sets of bars for *Epha2*; why is that?

These are EPHA2 peptides that are differentially phosphorylated, which we have now clearly indicated in Figure 1e.

3) Fig3D: the reference bar does not have an error bar, which should be included (this is also the case for several other plots).

The value obtained for the control (Time 0) in every experiment has been considered =1 in order to normalize and collate the data for three independent experiments.

4) Sox2 siRNA results are not in line with published data that show that loss of Sox2 leads to loss of Oct4 expression (Masui et al., 2007). This should be addressed.

We clearly show that efficient (~80%) Sox2 siRNA suppression disrupts *Nanog* expression. However, we cannot rule out the possibility that residual Sox2 expression may be sufficient to sustain *Oct4* expression. We have now addressed this caveat in the text.

5) Significance of Fgf5 and Brachyury upregulation is indicated (Fig3C), but is significance based on comparison with WT of with rescue cell lines?

This is significance resulting from a T-test comparing every group (knockout or rescued line) to WT control, which we have now clearly stated this in the figure legend.

6) Why is the duration of the timecourse different between the Western and IP-Western panels in Fig 3D. Is this a different experiment?

These were taken from different experiments, because we retrospectively examined EPHA2 tyrosine phosphorylation as a control to demonstrate effective EPHA2 activation by EFNA1 ligand. Nonetheless, we have included new data to show an equivalent time course.

7) Fgfr1 has recently been reported to be the main driver of Fgf/ERK signalling in ES Cells (Molotkov et al., 2017); therefore it is rather surprising that the authors detect an average of only < 5 molecules (maybe even less, this is hard to tell from the graph) per cell in Fig2a. Can they comment on that?

Our proteomics analysis calculates that mESCs express on average 2168 ± 42 copies of FGFR1 per cell (The y-axis is protein copies in thousands). This expression level is similar to other signaling receptors such as TGF-beta receptor and EGFR family members.

8) Fig2D appears to show an unspecific band, which should be indicated as such.

We thank the reviewer for pointing this out, and have now indicated the non-specific band with an asterisk.

Reviewers' comments:

Reviewer #1 (Remarks to the Author):

The authors have adequately addressed my concerns in the revision, and I am satisfied with the proposed changes. Especially, the new phosphoproteomics dataset in biological triplicates using TMT 9-plex isobaric labeling strongly improves the manuscript and the claims made. I have no further queries; I think the revised manuscript is acceptable for publication pending uploading of the raw MS files to the Proteome Exchange server.

Reviewer #2 (Remarks to the Author):

The authors have responded to my questions with additional experimental data which support their major conclusions. I therefore recommend accepting the revised manuscript without further revisions.

Reviewer #3 (Remarks to the Author):

The manuscript leaves this reviewer puzzled, even in its revised form. Clearly, there is most likely an impact of Ephrin signaling on ES cell self-renewal and differentiation. The authors fail however, to perform and present conclusive evidence for their hypotheses. The experiments performed in the revision are not sufficiently and unambiguously addressing my comments.

In summary, I believe that the authors have made an interesting discovery, but fail to show convincing data that support their claim of an interplay between Fgf/ERK and Ephrin signaling that regulates self-renewal and differentiation. The manuscript contains a large number of poorly controlled experiments (e.g. most Western blots do not have a loading control, see comments on commitment assay below), interpretations are often not in line with the actual data, and the authors miss an opportunity to contribute to our understanding of the signaling complexity involved in ES cell differentiation.

For these and the reasons outlined below, I cannot recommend this manuscript for publication in Nature Communications.

Relating to my previous points:

1) I remain unconvinced by the analysis of ES cell differentiation presented in the revised manuscript. I wonder to what extent the addition of EFNA1 is physiological and what exactly these data say about ES cell biology. The authors do actually perform a quite elegant experiment where they mix Epha2 KO;EFNA1 expressing cells with WT cells and observe induction of pTyr after EPH2 IP. This experiment is dedicated only a single panel in Fig2. It will be crucial to show that the key findings reported are not due to pre-clustering and/or unphysiological levels of EFNA1, but can be achieved in a more physiological setting.

b) In this regard I wonder to what extent the signaling cascade the authors describe is active in steady state ES cells, either cultured in FCS/LIF or in 2i, where Fgf signaling is reduced and Ephrin signaling should be more active (according to the authors' hypothesis). Figure3F shows the known response of ESCs to Mek inhibition, but unfortunately no information on Epha2 regulation and activation is given.

c) I do not understand the experiment which the authors claim to be a commitment assay. As far as I

understand the experiment as described, refers to a cell viability assay after replating in FCS/LIF after an initial phase of differentiation, whereas in a commitment assay the retention of ES colony forming potential should be measured. FCS/LIF medium is not selective for ES cells, therefore no conclusions can be drawn from this experiment.

d) As several thousand genes show differential expression between ES cells and early differentiation stages, correlated expression patterns are a poor predictor of functional interaction. In fact, the expression of Dnmt3b increases on day2 (FigS3E) concomitantly with an increase in Epha2 expression. In general, EB differentiation will not provide the temporal resolution to assess dynamics in regulation with accuracy. I do not understand why the authors did not profile expression patterns in the more controlled setup of Fig 4ij (defined medium conditions, presumably starting from a population of naïve pluripotent ES cells rather than FCS/LIF cultured heterogeneous cells). This is a missed opportunity.

2) Indeed the 5A-KI ES cell experiments are indicative of a role of Ephrin signaling in maintaining ESC identity. However, Fig4E shows a bright field image and Fig4F some Western blots (without loading control!). Nanog appears to be increased, Dnmt3b reduced. However, the authors leave it at that. Why are these ES cells not properly characterized? Is in these cells Fgf signaling reduced (as would be predicted)? The role of Ephrin signaling in ESC self-renewal is the key point this paper wants to address. In my opinion the authors fail to perform the appropriate experiments to test/falsify this hypothesis.

3) How does Fig3C show that "EPHA2 disruption interferes with pluripotency maintenance"? Both KO and WT cells show the same intensity of Nanog expression if not stimulated with EFNA1 (in fact the KO cells express a bit more, which is however difficult to assess because there is no loading control). I cannot share the enthusiasm for experiments in Fig3F. I have no doubt, that Epha2 interacts with Ptpn11, but this is by no means evidence for sequestering Shp2 away from Fgfr complexes and thereby negatively regulating Fgf/ERK signaling. In fact, the timings of (delayed, 30min after EFNA1 treatment) ppERK1/2 reduction and instant SHP2 "sequestration" are inconsistent with the author's hypothesis. It is disappointing that not even basic controls like a reciprocal CoIP (pull down Ephna2, probe for Shp2) have been performed.

The authors reference their previous publication to illustrate that Fgf4 addition to Fgf^{-/-} ESCs photocopies Epha2 depletion. There is however a discrepancy in that Nanog is depleted upon Fgf4 addition, whereas it remains unchanged in Epha2 KO cells. I understand that the experimental system might differ, but such a discrepancy makes it impossible to state that the effect of Fgf4 addition and Epha2 depletion are identical.

5) I stand by my statement that the link to the pluripotency network remains weak. The added experiments do not change that, as the conclusions are still based on Oct4 siRNA based KD, which will have profound effects on cell identity. As the authors have already shown previously in this manuscript that Ephna2 expression is linked to pluripotency, the fact that Oct4KO induced differentiation results in reduced Ephna2 expression is a self-fulfilling prophesy. A potential role of Nanog is mentioned but not tested.

Reviewers' comments:

Reviewer #1 (Remarks to the Author):

The authors have adequately addressed my concerns in the revision, and I am satisfied with the proposed changes. Especially, the new phosphoproteomics dataset in biological triplicates using TMT 9-plex isobaric labeling strongly improves the manuscript and the claims made.

I have no further queries; I think the revised manuscript is acceptable for publication pending uploading of the raw MS files to the Proteome Exchange server.

Reviewer #2 (Remarks to the Author):

The authors have responded to my questions with additional experimental data which support their major conclusions. I therefore recommend accepting the revised manuscript without further revisions.

Reviewer #3 (Remarks to the Author):

The manuscript leaves this reviewer puzzled, even in its revised form. Clearly, there is most likely an impact of Ephrin signaling on ES cell self-renewal and differentiation. The authors fail however, to perform and present conclusive evidence for their hypotheses. The experiments performed in the revision are not sufficiently and unambiguously addressing my comments.

In summary, I believe that the authors have made an interesting discovery, but fail to show convincing data that support their claim of an interplay between Fgf/ERK and Ephrin signaling that regulates self-renewal and differentiation. The manuscript contains a large number of poorly controlled experiments (e.g. most Western blots do not have a loading control, see comments on commitment assay below), interpretations are often not in line with the actual data, and the authors miss an opportunity to contribute to our understanding of the signaling complexity involved in ES cell differentiation.

We thank the referee for again acknowledging that our paper reports the interesting discovery of a new function for EPH-Ephrin signaling in regulating ES cell pluripotency and differentiation. We believe we have convincingly demonstrated functional interplay between FGF4-ERK1/2 and Ephrin signaling in ES cells, based on the following pieces of evidence;

- 1) FGF4-ERK1/2 signalling drives EPHA2 Ser/Thr phosphorylation at a motif, which when phosphorylates suppresses pluripotency gene expression and naïve pluripotent cell morphology.
- 2) FGF4-ERK1/2 signalling disrupts the pluripotency gene network and suppresses EPHA2 expression.
- 3) EPHA2 knockout promotes downregulation of pluripotency factors and differentiation.

I am surprised the referee suggests that the manuscript contains a “large number of poorly controlled experiments” and that “most western blots do not have a loading control”. I would like to stress that all western blots highlighted by the referee below also include a total ERK1/2 loading control. We are also unsure why the referee considers our commitment assay to be a viability assay. Replating in LIF/FCS is a standard assay to assess ES cell commitment to differentiation. Indeed, we have shown in a different experiment that EPHA2 does not influence cell viability, confirming that results from the commitment assay do not simply reflect measurement of viability.

For these and the reasons outlined below, I cannot recommend this manuscript for publication in Nature Communications.

Relating to my previous points:

1) I remain unconvinced by the analysis of ES cell differentiation presented in the revised manuscript. I wonder to what extent the addition of EFNA1 is physiological and what exactly these data say about ES cell biology. The authors do actually perform a quite elegant experiment where they mix Epha2 KO;EFNA1 expressing cells with WT cells and observe induction of pTyr after EPH2 IP. This experiment is dedicated only a single panel in Fig2. It will be crucial to show that the key findings reported are not due to pre-clustering and/or unphysiological levels of EFNA1, but can be achieved in a more physiological setting.

We agree that this is a key issue that we have addressed in this manuscript. In order to

allay concerns that clustered EFN activation of EPHA2 is unphysiological, pluripotency marker, commitment and differentiation assays (Fig. 3A, 3B, 3D) are all performed in the presence of endogenous EFN ligand stimulation. Furthermore, we provide evidence that EPHA2 is activated by endogenous EFN ligands in cultured mESCs (Fig. 2G). We also employ EFNA1 expressing cells, as suggested by the reviewer, to demonstrate that physiological EFN-EPH signalling promotes pluripotency factor expression (Fig. 3C). These data provide consistent evidence that endogenous/physiological EFN stimulation of EPHA2 plays a key role in promoting expression of pluripotency genes and suppressing differentiation.

b) In this regard I wonder to what extent the signaling cascade the authors describe is active in steady state ES cells, either cultured in FCS/LIF or in 2i, where Fgf signaling is reduced and Ephrin signaling should be more active (according to the authors' hypothesis). Figure 3F shows the known response of ESCs to Mek inhibition, but unfortunately no information on Epha2 regulation and activation is given.

Again, we agree that this is a key point of our paper, and our data confirm that the signaling cascade we describe is indeed active in steady state ES cells. As noted above, we show that EPHA2 is active in mESCs cultured in LIF/FCS (Fig. 2G). Furthermore, we have clearly demonstrated that EPHA2 Ser/Thr phosphorylation is tightly regulated by FGF4-ERK1/2 signalling in ES cells (Fig. 1C, 1D, 1E, 1F, 4B), and that EPHA2 Ser/Thr phosphorylation inhibits receptor activation (Fig. 4C, 4D, Supp. Fig. 4B). Unfortunately, it is not possible to directly compare EPHA2 activation in mESCs cultured in LIF/FCS, 2i or as embryoid bodies, because EPHA2 expression is also under the control of FGF4-ERK1/2 signalling (Supp. Fig. 4E, 4F, 4G). Therefore, EPHA2 levels are very different in each of these cell systems, which makes it impossible to determine relative levels of activation. However, we circumvent this problem by employing EPHA2 Ser/Thr phosphorylation site mutants, and reiterate that this provides compelling evidence that EPHA2 activation is directly controlled by FGF4-ERK1/2 signalling.

c) I do not understand the experiment which the authors claim to be a commitment assay. As far as I understand the experiment as described, refers to a cell viability assay after replating in FCS/LIF after an initial phase of differentiation, whereas in a commitment assay the retention of ES colony forming potential should be measured. FCS/LIF medium is not selective for ES cells, therefore no conclusions can be drawn from this experiment.

We are surprised that the referee considers this a cell viability assay, as this assay is adapted from a standard method used to measure ES cell commitment. Indeed, we have shown using alternative methods that EPHA2 has no effect on cell viability (Supp. Fig. 3B, 3C), confirming that the result from the commitment assay does not reflect altered cell viability. In addition, we show that EFNA1 stimulation of Epha2^{+/+} mESCs drives retention of naïve morphology with increased alkaline phosphatase staining after the initial phase of differentiation (Supp. Fig. 3D), indicating that EPHA2 signalling promotes the pluripotent phenotype.

d) As several thousand genes show differential expression between ES cells and early differentiation stages, correlated expression patterns are a poor predictor of functional interaction. In fact, the expression of Dnmt3b increases on day2 (FigS3E) concomitantly with an increase in Epha2 expression.

In general, EB differentiation will not provide the temporal resolution to assess dynamics in regulation with accuracy. I do not understand why the authors did not profile expression patterns in the more controlled setup of Fig 4ij (defined medium conditions, presumably

starting from a population of naïve pluripotent ES cells rather than FCS/LIF cultured heterogeneous cells). This is a missed opportunity.

Three-dimensional aggregates of ES cells have been widely used as a model to recapitulate development and to dissect the molecular networks that regulate ES cell differentiation. In this model, we have used hanging-drop aggregation (1200 cells/drop), which results in homogeneous embryoid bodies that differentiate in a very reproducible fashion to recapitulate early developmental processes. However, as the referee points out, we show that *Epha2* and *Efna1* expression are regulated in a similar manner in defined medium conditions of 2i in N2B27 (Fig 4I, 4J), confirming the relevance of our findings across ES cell differentiation systems.

2) Indeed the 5A-KI ES cell experiments are indicative of a role of Ephrin signaling in maintaining ESC identity. However, Fig4E shows a bright field image and Fig4F some Western blots (without loading control!). Nanog appears to be increased, Dnmt3b reduced. However, the authors leave it at that. Why are these ES cells not properly characterized? Is in these cells Fgf signaling reduced (as would be predicted)? The role of Ephrin signaling in ESC self-renewal is the key point this paper wants to address. In my opinion the authors fail to perform the appropriate experiments to test/falsify this hypothesis.

We agree with the reviewer that the key point of our paper is to investigate the role of Ephrin signalling in ES cells. To this end, we have characterised the function of the Ephrin-EPHA2 pathway in depth using a variety of biochemical and genetic approaches. We also demonstrate using EPHA2 5A-KI ES cells that EPHA2 Ser/Thr phosphorylation suppresses characteristics of naïve pluripotent ES cells, including domed colony morphology (Fig. 4E) and transcriptional programme (Fig. 4F). Furthermore, we show that expression of OCT4, a master regulator of pluripotency, is increased in EPHA2 5A-KI ES cells (Fig. 4F). These data confirm that multiple molecular and phenotypic hallmarks of pluripotency are enhanced in EPHA 5A-KI cells, and we modify our conclusions to state that “these data support the notion that S898 motif phosphorylation inhibits EPHA2 activation, and suppresses key morphological and transcriptional characteristics associated with pluripotent mESCs” (Line 262). Unfortunately, the referee is incorrect that Fig. 4F is “without loading control”, as total ERK1/2 expression is provided as a protein loading control.

3) How does Fig3C show that “EPHA2 disruption interferes with pluripotency maintenance”? Both KO and WT cells show the same intensity of Nanog expression if not stimulated with EFNA1 (in fact the KO cells express a bit more, which is however difficult to assess because there is no loading control).

We thank the referee for highlighting this opportunity to clarify this point. As showed previously, *Epha2*^{-/-} mESCs express similar levels of NANOG, DNMT3B and other pluripotency factors in the presence of LIF (Fig S3A). In Fig 3C, mESCs are also cultured in LIF, and levels of NANOG are therefore similar as expected. More importantly, Fig 3C shows that activation of EPHA2 signalling by EFNA1 increases expression of pluripotency markers including NANOG. Therefore, we modify our conclusions to state that “activation of EPHA2 by EFNA1 supports key morphological and transcriptional characteristics of pluripotency, thereby uncovering a novel function for EPH-EFN signalling in mESCs” (Line 172). Again, we provide total ERK1/2 as a loading control in Fig 3C.

I cannot share the enthusiasm for experiments in Fig3F. I have no doubt, that *Epha2* interacts with Ptpn11, but this is by no means evidence for sequestering Shp2 away from Fgfr complexes and thereby negatively regulating Fgf/ERK signaling. In fact, the timings of

(delayed, 30min after EFNA1 treatment) ppERK1/2 reduction and instant SHP2 “sequestration” are inconsistent with the author’s hypothesis. It is disappointing that not even basic controls like a reciprocal CoIP (pull down Ephna2, probe for Shp2) have been performed.

We thank the referee for the suggestion that the EPHA2-SHP2 interaction may have other functions in regulating the FGF4-ERK1/2 pathway. Indeed, SHP2 engagement by EPHA2 may sequester SHP2 and/or drive dephosphorylation of key pTyr sites for ERK1/2 activation. Thus, we modify our conclusions in the text as follows; “EPHA2 activation inhibits ERK1/2, either by sequestering the key scaffold SHP2 or by recruiting SHP2 to dephosphorylate key phosphotyrosine sites that are required for ERK1/2 activation” (Line 220).

Our data also show that the interaction between EPHA2 and SHP2 is highly specific, and exquisitely dependent on EFNA1 ligand stimulation. As there is no interaction between EPHA2 and SHP2 under basal conditions (Fig. 3F), this provides an excellent internal control to confirm the specificity of interaction.

Finally, the referee considers EPHA2 recruitment of SHP2 within 5mins and ERK1/2 inhibition within 30mins to be “inconsistent with our hypothesis”. However, we argue that the kinetics of the system dictate that either SHP2 recruitment or sequestration to EPHA2 should lead to dephosphorylation of ERK1/2 after a short delay. This is because a step-wise and time-dependent sequence of enzymatically-controlled events (tyrosine dephosphorylation, Ras GTPase inactivation and RAF-MEK kinase dephosphorylation and inactivation) must occur before ERK1/2 is finally dephosphorylated.

The authors reference their previous publication to illustrate that Fgf4 addition to Fgf-/- ESCs photocopies Epha2 depletion. There is however a discrepancy in that Nanog is depleted upon Fgf4 addition, whereas it remains unchanged in Epha2 KO cells. I understand that the experimental system might differ, but such a discrepancy makes it impossible to state that the effect of Fgf4 addition and Epha2 depletion are identical.

We thank the referee for pointing out this discrepancy, and highlight our data in Fig. 3C showing that EPHA2 activation by EFNA1 drives NANOG expression, indicating that, like FGF4, EPH-EFN signaling has the capacity to regulate NANOG expression. Nevertheless, we modify our conclusions as follows; “Our data indicate that EFN-EPH and FGF4-ERK1/2 have broadly opposing functions in regulating expression of pluripotency gene network components. However, it should be noted that NANOG expression is not altered in *Epha2*^{-/-} mESCs (Fig. 3a)” (Line 212).

5) I stand by my statement that the link to the pluripotency network remains weak. The added experiments do not change that, as the conclusions are still based on Oct4 siRNA based KD, which will have profound effects on cell identity. As the authors have already shown previously in this manuscript that Ephna2 expression is linked to pluripotency, the fact that Oct4KO induced differentiation results in reduced Ephna2 expression is a self-fulfilling prophesy. A potential role of Nanog is mentioned but not tested.

I am surprised that the referee considers that “the link to the pluripotency network remains weak”. Our data clearly demonstrate a strong correlation between OCT4 and EPHA2 expression, with several OCT4 binding sites identified in key regulatory elements of the EPHA2 gene promoter regions, suggesting direct regulation of EPHA2 by OCT4. We now provide further evidence that EPHA2 expression is not caused by a change in cell identity following OCT4 knockdown. Acute (24h) OCT4 siRNA depletion suppresses EPHA2

expression without altering expression of pluripotency genes such as NANOG and SOX2 (Supp. Fig. 5B), indicating that cells have not undergone a profound change in identity. Together, our data provide compelling evidence that OCT4 plays a direct role in regulating EPHA2 expression, rather than an indirect role based on changes in cell identity following OCT4 knockdown.

As suggested by the reviewer, we also explore the function of NANOG and SOX2 on EPHA2 expression. siRNA depletion of NANOG or SOX2 by siRNA has no effect on EPHA2 expression (Supp. Figs. 5B, D), again suggesting specific regulation of EPHA2 by OCT4. We also test the role of NANOG using the BRD4/BET bromodomain inhibitor JQ1, which inhibits expression of pluripotency factors including NANOG and OCT4. JQ1 treatment again confirms a correlation between expression of EPHA2 and OCT4, but not NANOG (Supp. Fig. 5C). However, we do not formally rule out the possibility that other pluripotency factors may have a functional role in EPHA2 expression. Indeed, our analysis of ChIP-SEQ data identifies *Epha2* promoter binding sites for at least 7 pluripotency network factors. We have now modified our conclusions in the text to this effect (Line 333).

Reviewers' comments:

Reviewer #3 (Remarks to the Author):

The authors have addressed some of my comments, and I can agree to some of the reasoning in their response. I can however not see a substantial change in the manuscript that makes me reassess my opinion. I can only state again that the manuscript contains beautiful biochemistry and the indicative data towards a role of segregated EPH-EFN expression in lineage separation is interesting. The functional data towards a role of Ephrin signalling in ES cell self-renewal and differentiation remains weak and what exactly could be the actual impact of Ephrin signalling is unclear to me. It can not be expected that the authors solve Ephrin signaling in ESCs in a single paper. But a basic state-of-the-art characterization of the presented ESC lines and conditions regarding self renewal and differentiation is a minimum requirement.

I want to clarify three issues, where I probably did not explain my point well enough:

-To the point regarding the commitment assay: The authors state to have performed the commitment assay according to Betschinger et al. Betschinger et al is in fact an ideal reference (and is widely used), but the authors have adapted the protocol to an extent where it no longer can provide the desired results. In a commitment assay, control cells will actually have to commit to differentiation. Therefore only few colonies should grow after re-plating in ESC conditions. ESC conditions ideally mean 2i medium, as in FCS/LIF multiple cell types will be able to survive, regardless of ES cell identity. In the data shown it appears as if in control cells a large number of densely growing AP positive ESC colonies grow after the commitment assay. In the absence of a 2i control where the same number of cells was plated and analysed data cannot be unambiguously interpreted. Showing a scan of the entire well of AP stained colonies would probably be a more ideal way of showing data here. This experiment as shown here is impossible to interpret in terms of commitment. The fact that the morphology changes upon EFNA1 addition in WT cells is interesting but not necessarily indicative of a commitment defect. The cell count in Figure 3b measures cell survival and not commitment. Images in Supplementary Fig3 are of little help for the interpretation. Therefore I uphold my statement that the authors have performed a cell viability assay (during a phase of differentiation), rather than a commitment assay.

-My comment regarding profiling of naïve markers in an exit from pluripotency assay (as in Figs 4i&j) was not intended to say that EBs are not an appropriate model system for differentiation, but that the suggested experiment would substantiate the statement that the exit from pluripotency is regulated by Ephrin signalling and complement the commitment assay.

-I agree that total ERK levels are a good control to assess the relative levels of ERK activation - experimental conditions used in this study might not affect total ERK levels (but can we be certain when signalling and cell state is perturbed?). The interpretation of results might however be a different one whether total ERK levels change or not.

I do not understand why for Figs 1f, 3a, 4b, 4f, 5a, S3a, S3F, s5a-d standard loading controls have not been used.

1. In the commitment assay, they feel that most control cells should have committed and only a few colonies should have formed. They were also troubled that AP staining indicated densely populated plates, even in the controls. Quantitation should be provided. Quantitation of AP+ colonies should be provided, not simply the number of cells.

We thank the referee for clarifying their concerns over the experimental design and interpretation of this assay. We have now performed the commitment assay to replicate as closely as possible the experimental conditions reported in Betschinger et al, 2013, and now provide compelling evidence that EPHA2 restricts mESC commitment to differentiation (Figure 3B). This experiment incorporates several key modifications suggested by the reviewer, including 2i conversion of *Epha2*^{+/+} and *Epha2*^{-/-} mESCs to ensure only ES cells survive after replating. 2i mESCs were then differentiated for 4 days in N2B27 media, and differentiating populations re-plated in 2i media and cultured for 3 days to expand surviving ES cell colonies. Alkaline phosphatase (AP) staining and quantification of the number of AP+ colonies clearly indicates a significant increase in mESC commitment of *Epha2*^{-/-} mESCs. Furthermore, we now include a scan of the entire well of AP stained colonies (Figure 3B), and a 2i control to unambiguously demonstrate that most mESCs had committed to differentiation during our assay (Supplementary Figure 3D).

2. In their experience conversion to 2i conditions takes only a few passages and experiments involving this conversion should only take a few weeks. Only cell lines central to conclusions of the manuscript would have to be converted.

We thank the referee for their input on this matter. Using 2i converted *Epha2*^{+/+} and *Epha2*^{-/-} mESCs as requested, we now provide further evidence that EPHA2 restricts exit from naïve pluripotency towards differentiation. Specifically, we show that EPHA2 suppresses expression of the neural marker *Sox1* during naïve pluripotent exit (Supplementary Figure 3G). This experiment substantiates the statement that exit from naïve pluripotency is regulated by Ephrin signalling and complements our results from the commitment assay and embryoid differentiation experiments.

Reviewers' comments:

Reviewer #3 (Remarks to the Author):

In the revised form the authors have performed some experiments to clarify the role of Ephrin signalling in pluripotency and naïve pluripotency exit.

In the new commitment assay the difference between Ephna2 KO and WT cells is clearly visible. This, together with data in Fig3a is a good indication that Epha2 KO ES cells differentiate faster.

I am however, still not certain, whether the assay performed can formally be called a commitment assay, because WT cells do hardly commit over a 96h period in Fig 3b. The expectation is that very few colonies survive (I have mentioned this before). In Fig 3b, 1000s of colonies are present. I can understand that the author's purpose was to see a reduced number of colonies in the mutant cells, therefore a higher baseline in the controls allows better quantification. But to allow proper quantitative interpretation, a lower percentage of cells after the 4 days of N2B27 exposure should have been plated.

In general, the commitment experiment and its quantification must be described better. I have the following points regarding the commitment assay:

- What exactly was quantified in Suppl Fig 3D? The images in Fig3b? If that is the case, then the quantification is rather pointless. Both in 2i and in N2B27, the plates are completely overgrown in WT controls. I assume (in the absence of information in the methods) the authors have multiplied the AP intensity by the factor 10 (I assume 2i (10%) means only 10% of 2i cultured cells were plated). If that is the case, then the quantification must be repeated, or data presented in a different manner. The image quality in Fig3b is so bad that I cannot be 100% sure about this, but to me it seems as if there was not difference between WT cells in 2i and N2B27. Both grow to completely fill up the plates. Therefore, it is not justified to plainly multiply AP intensity values by 10 (E.g. had the authors plated 50% of 2i cell, the result would most likely have been the same, because simply there cannot fit any more cells in the plate).

- I am surprised that the authors only measure a ~2-fold difference between Epha2 -/- and WT cells in Fig3b. The difference in the plates looks much more dramatic.

- Plots should be labelled AP density as this is what was measured, clonogenicity refers to the interpretation of these results. Ideally colonies should be counted, but that is impossible in such dense plates as shown.

- What is the unit in Suppl Fig 3D?

- As indicated, the image quality of Fig3B is suboptimal. A colour image is preferable to a black and white one to properly show the AP staining.

In response to my suggestion to repeat key experiments in 2i cultured cells, minimal analysis of 2i cultured cells have been performed and the authors now show increased Sox1 expression at d4 of differentiation in N2B27 medium in Epha2 KO cells compared to WT. The authors propose that this is indicative of a role for EPHA2 in suppression of lineage specific markers. With only one (neural) marker analysed I do not think that such a statement can be made. It could simply be a reflection of a delay in exiting naïve pluripotency. As mentioned before, not analysing a set of naïve marker genes

and early lineage markers in the 4d N2B27 time-course is a missed opportunity to clarify the exact role of Ephna2 in naïve pluripotency exit.

Reviewer #3 (Remarks to the Author):

In the revised form the authors have performed some experiments to clarify the role of Ephrin signalling in pluripotency and naïve pluripotency exit.

In the new commitment assay the difference between Ephna2 KO and WT cells is clearly visible. This, together with data in Fig3a is a good indication that Epha2 KO ES cells differentiate faster.

We thank the reviewer for their enthusiasm about the commitment assay, and agree that our data indicate that EPHA2 gene knock-out promotes ES cell differentiation. I am however, still not certain, whether the assay performed can formally be called a commitment assay, because WT cells do hardly commit over a 96h period in Fig 3b. The expectation is that very few colonies survive (I have mentioned this before). In Fig 3b, 1000s of colonies are present.

We apologise to the reviewer that the high level of ES cell commitment in our assay is not clearly represented in Fig 3b. Although we calculate the level of commitment to be at least 95% following 4 days of culture in N2B27 compared to 2i (Fig S3B), this was not apparent in Fig 3b because control and differentiating cells are plated at different densities and cultured for different times. In order to illustrate the high level of commitment more clearly, we provide new data in Fig 3b showing AP staining and quantification for equivalent replating of 2i and N2B27 ES cells in 2i (10% of the total in each case). As the referee correctly suggests, the vast majority of cells commit under these conditions, with few colonies surviving following N2B27 differentiation compared to 2i culture (<5%; Fig S3B). We also clarify our measurement of ES cell commitment in this assay with further detail in the figure legend and materials and methods.

I can understand that the author's purpose was to see a reduced number of colonies in the mutant cells, therefore a higher baseline in the controls allows better quantification. But to allow proper quantitative interpretation, a lower percentage of cells after the 4 days of N2B27 exposure should have been plated.

We provide new data in Fig 3b showing a lower percentage of cells plated following N2B27 exposure (10% N2B27 differentiated ES cells were replated for 4 days prior to AP staining, compared to 90% shown in the previous version). We also show that individual AP+ colonies are clearly visible, and that fewer AP+ colonies are recovered in EPHA2 knockout ES cells, demonstrating that EPHA2 knockout promotes ES cell commitment.

In general, the commitment experiment and its quantification must be described better.

We thank the reviewer for this helpful suggestion, and have now provided further experimental detail and quantification procedures for the commitment assay in the figure legends and materials and methods. For the avoidance of doubt regarding experimental procedure and quantification, ES cells were cultured in either 2i or N2B27 for 4 days, whereupon 10% of each cell population was replated in 2i. Of note, it was essential to perform AP staining of 2i control cells after 3 days in 2i and 3 days of replating, as plates were already very dense. AP staining of N2B27 exposed cells was performed after 4 days N2B27 differentiation and 3 days replating in 2i, as it proved impossible to quantify AP staining at earlier time points as the colonies are so small. This is clearly described in the methods and depicted in a modified schematic in Fig 3b.

I have the following points regarding the commitment assay:

- What exactly was quantified in Suppl Fig 3D? The images in Fig3b? If that is the case, then the quantification is rather pointless. Both in 2i and in N2B27, the plates are completely overgrown in WT controls. I assume (in the absence of information in the methods) the authors have multiplied the AP intensity by the factor 10 (I assume 2i (10%) means only 10% of 2i cultured cells were plated). If that is the case, then the quantification

must be repeated, or data presented in a different manner. The image quality in Fig3b is so bad that I cannot be 100% sure about this, but to me it seems as if there was not difference between WT cells in 2i and N2B27. Both grow to completely fill up the plates. Therefore, it is not justified to plainly multiply AP intensity values by 10 (E.g. had the authors plated 50% of 2i cell, the result would most likely have been the same, because simply there cannot fit any more cells in the plate).

We apologise for the confusion about how quantifications were performed, and clarify with new data and further experimental detail in the materials and methods. To summarise, Fig 3b now shows images of plates in which identical numbers of ES cells are replated in 2i following culture in either 2i control or N2B27 differentiation media. This analysis demonstrates that >95% of ES cells commit to differentiation in N2B27, confirming that the high level of commitment is not simply an artefact generated by factoring for dilution. We also apologise to the reviewer for the low image resolution, which was caused by file compression during the initial resubmission process, and now provide high-resolution colour images with zoom in Fig 3b (see high-resolution Fig 3 appended).

- I am surprised that the authors only measure a ~2-fold difference between Epha2 ^{-/-} and WT cells in Fig3b. The difference in the plates looks much more dramatic.

The difference in AP staining intensity is measured in triplicate using automated image quantification analysis. We also count AP⁺ colonies in each case, which shows a similar difference of ~2-3 fold between WT and EPHA2 knockout ES cells.

- Plots should be labelled AP density as this is what was measured, clonogenicity refers to the interpretation of these results. Ideally colonies should be counted, but that is impossible in such dense plates as shown.

We thank the reviewer for clarifying, and have now labelled the axes "AP staining intensity" rather than "clonogenicity". We also provide new data Fig 3b to demonstrate that individual AP⁺ colonies are clearly visible, and count individual colonies which confirms that the number of AP⁺ colonies is reduced in EPHA2 knockout ES cells.

- What is the unit in Suppl Fig 3D?

We thank the reviewer for pointing out this omission, and have modified the axis label to indicate "AP staining intensity"

- As indicated, the image quality of Fig3B is suboptimal. A colour image is preferable to a black and white one to properly show the AP staining.

We apologise to the reviewer for the low image resolution, which was caused by file compression during the initial resubmission process. We now provide high resolution colour images for Fig 3b.

In response to my suggestion to repeat key experiments in 2i cultured cells, minimal analysis of 2i cultured cells have been performed and the authors now show increased Sox1 expression at d4 of differentiation in N2B27 medium in Epha2 KO cells compared to WT. The authors propose that this is indicative of a role for EPHA2 in suppression of lineage specific markers. With only one (neural) marker analysed I do not think that such a statement can be made. It could simply be a reflection of a delay in exiting naïve pluripotency. As mentioned before, not analysing a set of naïve marker genes and early lineage markers in the 4d N2B27 time-course is a missed opportunity to clarify the exact role of Epha2 in naïve pluripotency exit.

We provide new data analysing suppression of pluripotency markers *Nanog*, *Klf4* and *Oct4* during N2B27 differentiation (Figure S3G). Consistent with our results from embryoid body differentiation assays (Fig 3a), EPHA2 knockout does not have a major impact on suppression of naïve pluripotency markers. However, EPHA2 gene knockout significantly increases induction of neural specific markers Sox1 and Nestin (Figure S3G), whilst the formative pluripotency/differentiation marker *Dnmt3b* and axonal marker *Kif1a* are also increased during N2B27 differentiation (Figure S3G). We also explore the function of

EPHA2 in mesendoderm differentiation using the 2i system. However, as reported previously, we show that expression of mesendoderm regulator *Brachyury* is high in 2i mESCs (Figure S3G) as a result of Wnt activation by the CHIR99021 GSK3 inhibitor. Therefore, whilst it is difficult to assess differentiation towards mesendodermal fate using 2i mESCs, our data from the embryoid body system (Fig 3a) shows that EPHA2 acts to restrict mesendoderm differentiation. In summary, our data indicating that EPHA2 suppresses neural lineage specific markers during 2i differentiation complements our embryoid body data showing that EPHA2 acts to suppress mesendoderm markers, and suggests that EPHA2 functions primarily to restrict expression of lineage specification factors during the initial stages of differentiation.

Figure 3b (top panel) high-resolution

REVIEWERS' COMMENTS:

Reviewer #3 (Remarks to the Author):

In the revised manuscript the commitment assay is now convincingly analysed and displayed. Differentiation defects in Epha2 KO cells can now be clearly appreciated. One major point remains to be addressed:

The authors have obtained additional data during monolayer N2B27 differentiation. This has led to the conclusion that: "...results indicate that EPHA2 suppresses induction of lineage specific markers to restrict mESC differentiation in distinct models. This is consistent with a general role of EPHA2 in regulating commitment to differentiation."

This is not as clear as portrayed by the authors and not entirely consistent with data shown.

Although it is shown that in EB differentiation Fgf5 and T are transiently increased in Epha2 KO cells, the situation in 2D differentiation is very different: Nestin is increased in 2i, and at d2; Sox1 only at day 4 and not before; and T shows an inverted pattern with lower expression in 2i in Epha2 KO ESCs. Unexpected T expression patterns are explained by expression in 2i induced by CHIRON. Although it is correct that T is already expressed in 2i, T expression is not reaching levels of mesendoderm progenitors, as shown by reporters and quantitative PCR (Mulas et al 2017). Therefore, loss of T repression upon Epha2 KO should still be readily detectable in a dynamic differentiation system (even in 2i). Data in Fig S3G are contrary to the authors model which would predict increased T expression in Epha2 KO cells.

In addition, the functional significance of rather small differences in lineage gene expression as portrayed in Fig. S3G are difficult to assess without a 'rescue' control.

Therefore, although the phenotype of Epha2 depletion is clear, the molecular function regarding lineage gene expression is not. This suggests a less specific role of Epha2 during differentiation where lack of Epha2 leads to generally aberrant differentiation, possibly by interfering with so far unknown mechanisms, rather than by specifically regulating lineage control genes. No direct interaction of Ephrin signalling and lineage gene expression has been shown. Data shown by the authors allow no strong conclusions about the molecular role of Epha2 in controlling lineage gene expression and should be portrayed and discussed accordingly.

Minor point:

On p9 the authors distinguish between statistically increased Sox1 increase and increased Kif1. I understand this in a way that Kif1 upregulation is not statistically significant. In my view data is either significant or, if not, the probability of error is too large to make a statement with reasonable certainty. I believe there is no justification for a distinction between 'increased' and 'increased with statistical significance'.

REVIEWERS' COMMENTS:

Reviewer #3 (Remarks to the Author):

In the revised manuscript the commitment assay is now convincingly analysed and displayed. Differentiation defects in Epha2 KO cells can now be clearly appreciated.

We appreciate the constructive feedback and are grateful to the referee for their enthusiasm for our improved analysis and display of the commitment assay. We agree that the function of EPHA2 in mESC commitment is now convincingly demonstrated.

One major point remains to be addressed:

The authors have obtained additional data during monolayer N2B27 differentiation. This has led to the conclusion that: "...results indicate that EPHA2 suppresses induction of lineage specific markers to restrict mESC differentiation in distinct models. This is consistent with a general role of EPHA2 in regulating commitment to differentiation."

This is not as clear as portrayed by the authors and not entirely consistent with data shown.

Although it is shown that in EB differentiation Fgf5 and T are transiently increased in Epha2 KO cells, the situation in 2D differentiation is very different: Nestin is increased in 2i, and at d2; Sox1 only at day 4 and not before; and T shows an inverted pattern with lower expression in 2i in Epha2 KO ESCs. Unexpected T expression patterns are explained by expression in 2i induced by CHIRON. Although it is correct that T is already expressed in 2i, T expression is not reaching levels of mesendoderm progenitors, as shown by reporters and quantitative PCR (Mulas et al 2017). Therefore, loss of T repression upon Epha2 KO should still be readily detectable in a dynamic differentiation system (even in 2i). Data in Fig S3G are contrary to the authors model which would predict increased T expression in Epha2 KO cells.

In addition, the functional significance of rather small differences in lineage gene expression as portrayed in Fig. S3G are difficult to assess without a 'rescue' control.

Therefore, although the phenotype of Epha2 depletion is clear, the molecular function regarding lineage gene expression is not. This suggests a less specific role of Epha2 during differentiation where lack of Epha2 leads to generally aberrant differentiation, possibly by interfering with so far unknown mechanisms, rather than by specifically regulating lineage control genes. No direct interaction of Ephrin signalling and lineage gene expression has been shown. Data shown by the authors allow no strong conclusions about the molecular role of Epha2 in controlling lineage gene expression and should be portrayed and discussed accordingly.

We again thank the referee for their constructive comments, and agree that although the role of EPHA2 during mESC commitment to differentiation is clear, the function of EPHA2 is regulating lineage-specific commitment and gene expression should be further investigated in future. We have now emphasised this important point in the results (p8 line 24 – p9 line 22) and discussion (p16 line 11-12).

Minor point:

On p9 the authors distinguish between statistically increased Sox1 increase and increased Kif1. I understand this in a way that Kif1 upregulation is not statistically significant. In my view data is either significant or, if not, the probability of error is too large to make a statement with reasonable certainty. I believe there is no justification for a distinction between 'increased' and 'increased with statistical significance'.

We thank the referee for raising this important point. We have now revised the text to state that whilst Kif1a expression in Epha2^{-/-} mESCs shows an upward trend, the increase is not statistically significant (p9 line 15).